# BEYOND DAGS: A LATENT PARTIAL CAUSAL MODEL FOR MULTIMODAL LEARNING

## ABSTRACT

Directed acyclic graphs (DAGs) are often assumed in causal discovery, however, accurately identifying these DAGs necessitates various assumptions, particularly in latent causal models, which can be challenging to validate in real-world applications. This raises a critical question: Are DAG assumptions truly necessary for certain applications? In this work, we introduce a novel latent partial causal model for multimodal data, which features two latent coupled variables, connected by an undirected edge, effectively representing transferable knowledge across different modalities. We focus on a prominent learning framework, e.g., multimodal contrastive learning, and demonstrate that, with certain statistical assumptions, multimodal contrastive learning successfully identifies the latent coupled variables up to trivial transformation. This finding enhances our understanding of the mechanisms driving the success of multimodal contrastive learning. Furthermore, this finding reveals a unique potential for disentanglement in multimodal contrastive representation learning, improving the utility of pre-trained models like CLIP that are trained using this approach. Through experiments with synthetic data, we demonstrate the robustness of our findings, even in the presence of violated assumptions. In addition, we validate the disentanglement capabilities of pre-trained CLIP in learning disentangled representations, facilitating few-shot learning and improving domain generalization across a diverse range of real-world datasets.

## 1 INTRODUCTION

The assumption of directed acyclic graphs (DAGs) in causality generally serves as a foundational principle that simplifies the representation and analysis of causal relationships, enhancing our understanding and estimation of causal effects across various domains (Pearl, 2000; Spirtes et al., 2001). To identify such DAGs, several assumptions are necessary to enforce the asymmetry between cause and effect nodes, as a result, making it possible to determine a unique causal direction. Typically, for causal discovery in observed space, these include constraints on the function class, such as the linear non-Gaussian assumption (Shimizu et al., 2006) or additive noise models (Hoyer et al., 2008; Peters et al., 2014). However, given the unknown and complex nature of real-world applications, it is challenging to justify these restricted function classes. This challenge is especially pronounced in latent space, such as in causal representation learning (Schölkopf et al., 2021), one of the most prominent subfields of causality. It seeks to uncover high-level latent causal variables and their corresponding DAGs using only observational data. To identify causal representations and the associated DAGs, most existing works require sufficient changes in the latent causal variables to ensure data availability resulting from interventions on all latent variables (Brehmer et al., 2022; Buchholz et al., 2023; Varici et al., 2023; Ahuja et al., 2023; Seigal et al., 2022; Liu et al., 2022; 2024b;a). However, acquiring such interventional data with sufficient changes can be quite challenging. This raises a natural question: Are DAG assumptions truly necessary for certain applications?

To address the question mentioned above, we focus on the multimodal data generative process, as it is in general difficult to determine, at first glance, which modality, such as text or image, serves as the cause and which as the effect. To formulate the causal generative process, we adopt a causal representation learning perspective by focusing on high-level latent causal factors, rather than low-level observed data such as image pixels or text words. Specifically, we propose a novel latent partial causal model for the generative process of multimodal data, as illustrated in Figure 1. Rather than relying on the traditional DAG assumption, this framework introduce a couple of latent variables (i.e.,

latent coupled variables), connected by an undirected edge. This structure provides greater flexibility for modeling transferable knowledge across modalities, as it allows for multiple possibilities, as illustrated in depicted by Figure 2. For instance, it can adapt to models by enforcing an identical mapping on the undirected edge between the latent coupled variables, or it can accommodate a latent confounder influencing both coupled variables. Additionally, the proposed model incorporates modality-specific latent variables to capture unique information within each modality and employs distinct mappings from the latent space to the observed space to generate data for different modalities.

Given the proposed latent coupled model to model *generative* process, we analyze it within the widely-used multimodal contrastive learning (e.g., *inference*) paradigm (Zhang et al., 2022b; Radford et al., 2021), which has demonstrated significant potential across various downstream tasks. Specifically, we parameterize the proposed latent causal generative model according to two distinct types of latent spaces: hypersphere and convex bodies, respectively. Our analysis show that multimodal contrastive learning can identify latent coupled variables up to a trivial linear transformation in hypersphere space, and up to permutation transformation in convex bodies, respectively. These results demonstrate that the learned representations by multimodal contrastive learning capture essential latent coupled variables within the data, while suppressing irrelevant variant part, e.g., modality-specific latent variables. Consequently, our theoretical findings provide a solid foundation for the success of multimodal contrastive learning. Beyond understanding the success of multimodal contrastive learning, our theoretical results also unlock their disentanglement potential. Importantly, the emergence of the disentanglement ability holds significant potential for promoting the utilization of pre-trained models, such as CLIP (Radford et al., 2021), trained by multimodal contrastive learning. guided by the identifiability result in hypersphere space that indicates the existence of a linear transformation, we can leverage linear independent component analysis (ICA) (Hyvärinen et al., 2001) to unlock the benefits of CLIP-like models across various downstream tasks, including but not limited to: learning disentangled representations, enhancing few-shot learning, and improving domain generalization. We validate our theoretical findings under ideal conditions and demonstrate their robustness even with partial violations of assumptions. Experiments on real datasets show that coupling with ICA effectively enhances the potential of pre-trained CLIP-based methods for various downstream tasks.

In summary, our contributions are as follows:

- We propose a novel latent partial causal model for the generative process, specifically designed for multimodal data. Instead of DAGs, our model introduces latent coupled variables, connected by undirected edges, to capture transferable knowledge across modalities.

- Our analysis shows that multimodal contrastive learning can identify latent coupled variables within the proposed model, providing a solid foundation for its success.

- Beyond explaining the success of multimodal contrastive learning, to the best of our knowledge, this is the first work to provide guarantee for the potential for disentanglement, pushing the boundaries of how pre-trained models, *e.g.*, CLIP, can be utilized (Radford et al., 2021).

- We validate our theoretical findings under ideal conditions and demonstrate their robustness even when some assumptions are partially violated. Extensive experiments across various tasks, including few-shot learning, domain generalization, and disentangled representation learning on over 16 real-world datasets, support the effectiveness of latent coupled models.

## 2 RELATED WORK

**Multimodal contrastive representation learning**   Multi-modal contrastive representation learning, driven by underlying transferable knowledge across modalities, aims to coalesce inputs from these diverse sources into a cohesive representation space. This is typically achieved using a symmetric version of the standard contrastive loss (Oord et al., 2018; Gutmann and Hyvärinen, 2010), a method designed to align accurate pairings while distinguishing incorrect ones (He and Peng, 2017; Radford et al., 2021). Although this approach has proven successful in a range of downstream tasks (Radford et al., 2021; Zhou et al., 2022a;b; Lüddecke and Ecker, 2022; Ban and Dong, 2022), there remains a gap in our comprehensive theoretical and empirical understanding of the representations it learns. Recently, there has been a growing interest in exploring multi-modal contrastive learning from various perspectives. For instance, the study by Liang et al. (2022) provides insights into the modality gap

inherent in multi-modal contrastive learning. Similarly, the research presented by Nakada et al. (2023) establishes a link between general multimodal contrastive loss and SVD analysis. Additionally, Huang et al. (2021) posits that learning with multiple modalities can lead to a reduced population risk compared to using a subset of these modalities. Diverging from these approaches, our work delves into multi-modal contrastive representation learning by examining its connection with generative models.

Past research has sought to comprehend the representations derived from standard single-modality contrastive learning, examining them through the lens of alignment and uniformity (Wang and Isola, 2020), showing guarantees on the performance of the learned representations on the average classification task (Saunshi et al., 2019), or in terms of the identifiability of latent variables (Zimmermann et al., 2021; Von Kügelgen et al., 2021). Building on these foundations, our work takes a foreword step. We demonstrate that multi-modal contrastive learning can identify latent coupled variables, extending the insights from previous studies into the realm of multi-modality.

Very recently, several studies have emerged, focusing on multi-modal settings (Daunhawer et al., 2023; Yao et al., 2023). A clear distinction is that: the proposed model captures transferable knowledge across modalities by an undirected edge between latent coupled variables, while previous works often achieve it by introducing shared variables (Daunhawer et al., 2023; Yao et al., 2023). Notably, our modeling approach is more general, as it can be reduced to the shared variables used in previous works (Daunhawer et al., 2023; Yao et al., 2023) by enforcing an identical mapping on the undirected edge between latent coupled variables. Some of these works have only achieved partial identifiability of coupled variables (Daunhawer et al., 2023; Yao et al., 2023), specifically identifying latent content variables but not latent style variables. In contrast, our work achieves comprehensive identifiability results for all latent coupled variables, offering a deeper level of understanding. Our research also diverges from the approach taken in Gresele et al. (2020) in two key ways: Firstly, we model ransferable knowledge across modalities using conditional distributions, whereas the latter utilizes identical variables for this purpose. Secondly, while Gresele et al. (2020) relies on the premise that the mapping from the latent space to observations must be constrained by component-wise corrupters to ensure identifiability, our findings do not necessitate such constraints.

**Nonlinear ICA**  Nonlinear Independent Component Analysis (ICA) aims to unravel latent independent variables from observational data that has been subject to a nonlinear mixture of these latent factors. However, as pointed out in the seminal work by Hyvärinen and Pajunen (1999), solving this problem is generally infeasible without specific underlying assumptions. A prominent direction in contemporary research leverages the concept of distributional changes in latent variables, which leads to the creation of multi-domain observational data. This approach has been extensively explored and developed in a series of studies (Hyvarinen and Morioka, 2016; 2017; Hyvarinen et al., 2019; Khemakhem et al., 2020), each contributing to a deeper understanding and more refined methodologies in the field of Nonlinear ICA. We build upon this body of research by incorporating co-occurrence patterns observed across multiple modalities. It is important to note the distinct difference between multi-domain and multi-modal approaches. The former typically implies a consistent mapping from the latent space to the observational space across all domains, whereas the latter accommodates different mappings for each modality. Additionally, while multi-domain approaches generally assume a totally shared latent variables across all domains, multi-modal methods allow for the existence of modality-specific latent variables.

## 3 THE PROPOSED LATENT PARTIAL CAUSAL MODELS AND INTUITION

In this section, we introduce a novel latent partial causal model to represent the generative process for multimodal data. Unlike traditional DAG assumptions, our model allows for an undirected edge between two variables to capture transferable knowledge across modalities as depicted by Figure 1. This undirected structure allows for the representation of multiple DAG assumptions as depicted by Figure 2, enhancing flexibility in modeling transferable knowledge. Consequently, it enables the extraction of common knowledge that summarizes the underlying principles behind various DAG assumptions. Building on this, we offer preliminary insights within the multimodal contrastive learning framework, demonstrating that it provides two fundamental factors for solving inverse problems: prior matching and information preservation.

### 3.1 THE PROPOSED LATENT PARTIAL CAUSAL MODELS

Figure 1 illustrates the proposed latent partial causal model. In this model, the whole latent space is partitioned into two parts, each representing a different modality, *e.g.*, image and text. More specifically, to model transferable knowledge across modalities, an undirected edge is established between latent coupled variables, $z_x$ and $z_t$. The rationale behind this modeling approach is grounded in the recognition that real-world multimodal data is often complex, noisy, and multifaceted. On one hand, the assertion 'a picture is worth a thousand words' is well-supported in literature (Gropper, 1963; Hum et al., 2011), emphasizing the rich detail and information that images can convey compared to text. Conversely, this notion is not universally applicable as argued by Reinert (1976), which suggests

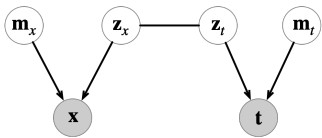

Figure 1: The proposed latent partial causal model, where $z_x$ and $z_t$ denote latent coupled variables, and $m_x$ and $m_t$ denote modality-specific latent variables. The observations $x$ (e.g., images) and $t$ (e.g., text) are generated by two distinct generative processes, respectively.

that sometimes textual information can be more informative than visual data. This perspective is further echoed by Fidler et al. (2013) in their assertion that 'a sentence is worth a thousand pixels', highlighting the potential of text in conveying complex ideas succinctly. In addition, we introduce $m_x$ and $m_t$ modality-specific latent variables, each tailored to capture the unique characteristics of their respective domains. For example, $m_x$ could encode information focusing on aspects like the presence of background noise or other visual artifacts that contribute to the overall composition of an image. On the other hand, $m_t$ could encode information about sentence structure or linguistic patterns that are characteristic of the grammar in textual content. Finally, the observations are associated with two distinct generative processes. Specifically, $x$ (*e.g.*, images) are generated through the process $g_x(m_x, z_x)$, while observation $t$ (*e.g.*, text) come into existence through the process $g_t(m_t, z_t)$.

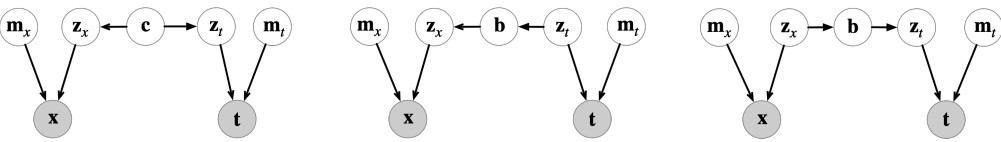

Figure 2: Illustrative DAGs behind the proposed partial causal model. From left to right, a latent confounder influences both $z_x$ and $z_t$. In the second structure, $z_t$ leads to an intermediate mediator node $b$, which in turn influences $z_x$. Here, mediator $b$ plays a bottleneck role in determining transferable knowledge. The inverse relationship is illustrated symmetrically in the right DAG.

**Possible DAGs Behind the Proposed Model**   Figure 2 depicts potential DAGs underlying the proposed latent coupled models. Note that this representation does not cover all possible DAG configurations. For the left DAG in Figure 2, the latent confounder $c$ could be understood as a hidden variable that influences both $z_x$ and $z_t$, In other words, $c$ is a shared source of variation between two modalities. This confounder could represent some underlying concept or context that ties the image and text together. For instance, if both the image and text are about "sports," the confounder might capture the general topic of sports. For the middle DAG, In this context, $b$ could represent the transferable knowledge between the two modalities (e.g., text and image). It may capture a more general or abstract concept that is derived from the latent variables $z_t$ and is used to inform the image latent space $z_x$. Moreover, considering $z_t$ as a ancestor node of $z_x$ is illustrated in the context of text-to-image ~~retrieval~~ generalization. In this scenario, the text query $z_t$ determines the features of the images, e.g., $z_x$. For the right subfigure, it can be conceptualized as an image captioning task. In this context, $z_x$ represents the latent features of the image, while $z_t$ corresponds to the latent representation of the generated caption. Here, again, $b$ serves as the bridge between the visual and textual modalities, encapsulating the transferable knowledge necessary to accurately convey the image's content in words.

In general, when considering the assumptions of any of these DAGs mentioned above, we often require various additional assumptions to fully identify the latent variables and their corresponding structures. For instance, many existing assumptions dictate that all latent variables must exhibit

sufficient variability, related to interventional data (Brehmer et al., 2022; Buchholz et al., 2023; Varici et al., 2023; Ahuja et al., 2023; Seigal et al., 2022; Liu et al., 2022; 2024b;a). However, validating this approach in real-world applications can be quite challenging. Acquiring such diverse data from different environments presents significant difficulties. Given this, we do not focus on the traditional problem setting in causal representation learning, which often assumes a directed acyclic graph (DAG) structure. Instead, we turn our attention to investigating the question: Can we derive advantages from non-DAG assumptions for certain applications, such as the proposed latent partial causal models?

## 3.2 INTUITION: PRIOR MATCHING AND INFORMATION PRESERVATION

Given the proposed latent coupled generative models, we consider an inference framework, multi-modal contrastive learning, to provide further analysis, as recent progress in this field indicate that representations learned through a multimodal contrastive learning are highly effective in various downstream tasks (Radford et al., 2021). The contrastive loss function is designed to maximize the similarity in the embedding space between modalities for real pairs, while minimizing the similarity for incorrect pairs. Formally, the optimization objective is as follows (Zhang et al., 2022b; Radford et al., 2021):

$$\mathcal{L} = -\frac{1}{N} \sum_{i=1}^{N} \log \frac{e^{-d\left(\mathbf{f}_x(\mathbf{x}_i), \mathbf{f}_t(\mathbf{t}_i)\right)/\tau}}{\sum_{j=1}^{N} e^{-d\left(\mathbf{f}_x(\mathbf{x}_i), \mathbf{f}_t(\mathbf{t}_j)\right)/\tau}} - \frac{1}{N} \sum_{i=1}^{N} \log \frac{e^{-d\left(\mathbf{f}_x(\mathbf{x}_i), \mathbf{f}_t(\mathbf{t}_i)\right)/\tau}}{\sum_{j=1}^{N} e^{-d\left(\mathbf{f}_x(\mathbf{x}_j), \mathbf{f}_t(\mathbf{t}_i)\right)/\tau}}, \quad (1)$$

where $d$ denotes a distance metric, *e.g.*, cosine similarity on hypersphere or L1 norm on convex bodies, $\tau$ is a learnable temperature parameter, $N$ denotes the sample size, which means that we have $N$ positive pairs and $N^2 - N$ negative pairs, $\mathbf{f}_x$ denote the encoder on one modality $\mathbf{x}$, *e.g.*, image, similarly, $\mathbf{f}_t$ denote the encoder on another $\mathbf{t}$, *e.g.*, text. To further understand the multimodal contrastive loss, we begin by investigating its asymptotics:

**Theorem 3.1** (Asymptotics of $\mathcal{L}$). *For fixed $\tau > 0$, as the sample size $N \to \infty$, the (normalized) multimodal contrastive loss converges to* [1]

$$\lim_{N \to \infty} \mathcal{L} - 2 \log N = 2 \underset{(\mathbf{x},\mathbf{t}) \sim p(\mathbf{x},\mathbf{t})}{\mathbb{E}} \left[ d\left(\mathbf{f}_x(\mathbf{x}), \mathbf{f}_t(\mathbf{t})\right)/\tau \right] + \underset{\mathbf{x} \sim p(\mathbf{x})}{\mathbb{E}} \left[ \log \underset{\mathbf{t} \sim p(\mathbf{t})}{\mathbb{E}} \left[ e^{-d\left(\mathbf{f}_x(\mathbf{x}), \mathbf{f}_t(\mathbf{t})\right)/\tau} \right] \right]$$

$$+ \underset{\mathbf{t} \sim p(\mathbf{t})}{\mathbb{E}} \left[ \log \underset{\mathbf{x} \sim p(\mathbf{x})}{\mathbb{E}} \left[ e^{-d\left(\mathbf{f}_x(\mathbf{x}), \mathbf{f}_t(\mathbf{t})\right)/\tau} \right] \right], \quad (2)$$

See Appendix A.1 for proof.

**Intuition** Our primary insight is that the loss function in Eq. (2) is intricately linked to two fundamental elements that are crucial for solving inverse problem, *i.e.*, identifying latent independent variables from observed data in nonlinear ICA:

- *Prior Matching*: The solution space is constrained by prior knowledge, which helps mitigate issues of non-uniqueness in identifying latent variables.
- *Information Preservation*: This ensures that the solution space can capture the full range and complexity of the latent variables derived from observed data.

To detail this, we first provide the following analysis:

**Prior Matching** To clarify this point, let us focus on the first term at the right-hand side of Eq. (2). In a multi-model setting, one modality plays a crucial role as a supervised signal for another modality. This implies that one modality can serve as potential prior knowledge. By minimizing the first term, which reduces the distance between features obtained by encoders for real pairs, we essentially ensure the features generated by one encoder, *e.g.*, on the image modality, closely approximate the prior knowledge provided by another modality, *e.g.*, text.

---

[1]This is a generalized version of the existing Theorem 1 in Wang and Isola (2020), specifically adapted to the multi-modality setting.

**Information Preservation**   We now focus on the last two terms on the right-hand side of Eq. (2). Essentially, these two terms can be approximated by optimizing the following expression (a proof can be found in Appendix A.2):

$$-H\big(p(\mathbf{f}_x(\mathbf{x})), p(\mathbf{f}_t(\mathbf{t}))\big) - H\big(p(\mathbf{f}_t(\mathbf{t})), p(\mathbf{f}_x(\mathbf{x}))\big), \qquad (3)$$

Here, $H(\cdot, \cdot)$ represents the cross entropy. Overall, the form of the objective function Eq. 2 is symmetric between $\mathbf{x}$ and $\mathbf{t}$. When searching for the minimum of the objective function, if $p(\mathbf{f}_x(\mathbf{x}))$ and $p(\mathbf{f}_t(\mathbf{t}))$ are not equal, the optimal solution may deviate, leading to an increase in the value of the objective function. In particular, the last two terms on the right-hand side may become asymmetric. Thus, to ensure reaching the optimal solution, the two distributions might be equal. In this context, the cross entropy in Eq. (3) will reduce to entropy. As a result, when both $\mathbf{f}_x$ and $\mathbf{f}_t$ transform $\mathbf{x}$ and $\mathbf{t}$ into uniformly distributed random variables, respectively, Eq. (3) reaches its optimal solution. This uniform distribution underscores our goal of finding functions $\mathbf{f}_x$ and $\mathbf{f}_t$ that maximize information preservation.

~~Expanding upon the intuition presented,~~ Prior works have investigated contrastive loss primarily in the context of single modality, focusing on two main perspectives: 1) the alignment-uniformity perspective Wang and Isola (2020), e.g., similar to prior matching and 2) the information preservation perspective Oord et al. (2018). However, these insights have largely been discussed separately. In this work, we present a novel insight that combines these two perspectives within the framework of solving inverse problems. Consequently, we posit that multimodal contrastive representation learning has the potential to identify latent variables in the proposed causal generative model. In the following section, we will parameterize the proposed latent partial causal model for further exploration.

## 4 IDENTIFIABILITY ANALYSIS ON HYPERSPHERE AND CONVEX BODIES

In this section, we conduct an identifiability analysis for the proposed latent partial causal model illustrated in Figure 1. Our analysis specifically focuses on two distinct types of latent spaces: hypersphere and convex bodies, which are explored under certain defined assumptions.

### 4.1 IDENTIFIABILITY ANALYSIS ON HYPERSPHERE

On hypersphere, we parameterize the proposed latent partial causal generative models depicted in Figure 1 by the following:

$$p(\mathbf{z}_x) = |\mathcal{Z}|^{-1}, \quad p(\mathbf{z}_t|\mathbf{z}_x) = C_p^{-1} e^{(k\mathbf{z}_t^T \mathbf{z}_x)}, \quad \mathbf{x} = \mathbf{g}_x(\mathbf{z}_x, \mathbf{m}_x), \quad \mathbf{t} = \mathbf{g}_t(\mathbf{z}_t, \mathbf{m}_t), \qquad (4)$$

where $\mathcal{Z}$ denotes the space of latent factors $\mathbf{z}_x$ and $\mathbf{z}_t$. Influenced by the commonly used feature normalization in constrastive loss, we assume that $\mathcal{Z}$ is the unit hypersphere $\mathbb{S}^{M-1}$. We do not enforce any further assumptions for $\mathbf{m}_x$ and $\mathbf{m}_t$. For $\mathbf{g}_x$ and $\mathbf{g}_t$, we assume them to be invertible (*i.e.*, injective) mapping, ensuring the information in latent space can be recovered. In addition, we assume that $p(\mathbf{z}_x)$ follows a uniform distribution, and $p(\mathbf{z}_t|\mathbf{z}_x)$ follows a von Mises-Fisher (vMF) distribution, considering the constraint of unit hypersphere. Given these assumptions, our subsequent discussion aims to establish that the minimization of the multimodal contrastive loss (as defined in Eq. (2)) converges to a symmetric cross entropy, as follows:

**Theorem 4.1.** *($\mathcal{L}$ converges to the symmetric cross-entropy) Under the assumptions defined in Eqs. (4) for the proposed latent partial causal model, the necessary condition $\mathbf{f}_x \circ \mathbf{g}_x = \mathbf{f}_t \circ \mathbf{g}_t$, denoted as $\mathbf{h}$, for the optimal normalized multimodal contrastiveloss given by Eq. (2) leads to the following reduction of the loss itself:* [2]

$$\lim_{N \to \infty} \mathcal{L} - 2\log N + 2\log |\mathcal{Z}| =$$

$$\mathbb{E}_{\mathbf{z}_x \sim p(\mathbf{z}_x)} \big[ H(p(\mathbf{z}_t|\mathbf{z}_x), q_{\mathbf{h}}(\mathbf{z}_t|\mathbf{z}_x)) \big] + \mathbb{E}_{\mathbf{z}_t \sim p(\mathbf{z}_t)} \big[ H(p(\mathbf{z}_x|\mathbf{z}_t), q_{\mathbf{h}}(\mathbf{z}_x|\mathbf{z}_t)) \big], \qquad (5)$$

*where $H$ is the cross entropy, the conditional distributions $q_{\mathbf{h}}(\mathbf{z}_t|\mathbf{z}_x)$ and $q(\mathbf{z}_x|\mathbf{z}_t)$ are parameterized by the following:*

$$q_{\mathbf{h}}(\mathbf{z}_x|\mathbf{z}_t) = C_q(\mathbf{z}_t)^{-1} e^{(\mathbf{h}(\mathbf{z}_x)^T \mathbf{h}(\mathbf{z}_t)/\tau)}, q_{\mathbf{h}}(\mathbf{z}_t|\mathbf{z}_x) = C_q(\mathbf{z}_x)^{-1} e^{(\mathbf{h}(\mathbf{z}_t)^T \mathbf{h}(\mathbf{z}_x)/\tau)}, \qquad (6)$$

---

[2]Theorem 4.1 is a generalization of finding outlined in Theorem 1 in Zimmermann et al. (2021) in the context of multi-modal setting.

*with*

$$C_q(\mathbf{z}_t) = \int e^{(\mathbf{h}(\mathbf{z}_x)^T \mathbf{h}(\mathbf{z}_t)/\tau)} \mathrm{d}\mathbf{z}_x, C_q(\mathbf{z}_x) = \int e^{(\mathbf{h}(\mathbf{z}_x)^T \mathbf{h}(\mathbf{z}_t)/\tau)} \mathrm{d}\mathbf{z}_t.$$

Refer to Appendix A.3.1 for proof.

~~**Bridging Multimodal Contrastive Loss with Single-Modal Contrastive Loss on Hypersphere**~~
By addressing various asymmetrical challenges arising from modality differences, such as modality-specific variables $\mathbf{m}_x$ and $\mathbf{m}_t$, as well as distinct generative processes $\mathbf{g}_x$ and $\mathbf{g}_t$, we originally develop the result in Theorem 4.1. This result establishes a crucial connection that bridges multimodal contrastive loss with traditional single-modal contrastive loss. This connection is particularly valuable as it enables the transfer of previously developed results from single-modal settings to the multimodal context, as follows: ~~By leveraging Theorem 4.1, the minimization of Eq. (5) identifies the latent variables $\mathbf{z}_x$ (symmetrically, $\mathbf{z}_t$) up to a linear transformation, *i.e.*, the recovered latent variable $\hat{\mathbf{z}}_x$, obtained through the minimization of Eq. (5), is linearly related to the true $\mathbf{z}_x$ as follows: $\hat{\mathbf{z}}_x = \mathbf{A}\mathbf{z}_x + \mathbf{c}$, where $\mathbf{A}$ represents an orthogonal matrix, and $\mathbf{c}$ is a constant vector.~~

**Corollary 4.2.** *By leveraging Theorem 4.1, the minimization of Eq. (5) identifies the latent variables $\mathbf{z}_x$ (symmetrically, $\mathbf{z}_t$) up to a linear transformation, i.e., the recovered latent variable $\hat{\mathbf{z}}_x$, obtained through the minimization of Eq. (5), is linearly related to the true $\mathbf{z}_x$ as follows: $\hat{\mathbf{z}}_x = \mathbf{A}\mathbf{z}_x + \mathbf{c}$, where $\mathbf{A}$ represents an orthogonal matrix, and $\mathbf{c}$ is a constant vector.*

See details in Appendix A.3.2.

~~**Insights** Thanks to the alignment between the assumptions in Theorem 4.1 and the training configuration in multimodal contrastive learning, the identifiability result mentioned above offers two key insights: 1) multimodal contrastive learning identifies latent coupled variables within the proposed partial causal model, providing a solid foundation for its success; and 2) Pre-trained model, such as CLIP, by multimodal contrastive learning, has the potential for disentanglement, which can be achieved by solving the linear transformation arising from the identifiability results, e.g., through linear ICA methods. A key factor is the hypersphere's geometry—specifically, the unit $M-1$ dimensional hypersphere—where the maximum number of independent dimensions is $M-1$. As a result, $\mathbf{z}_x$ can have at most $M-1$ independent dimensions. Even in cases where the latent variables are fully dependent, linear ICA may still work by extracting the most independent components using the principle of maximal non-Gaussianity (Hyvärinen et al., 2001). In our implementation, we use the FastICA algorithm from (Hyvarinen, 1999).~~

**Connection with disentanglement** Corollary 4.2 has shown that the minimization of Eq. (5) identifies the latent variables $\mathbf{z}_x$ (symmetrically, $\mathbf{z}_t$) up to a linear transformation. Note that the key difference between Eq. (5) and the multimodal contrastive loss lies in a constant term, specifically $-2\log N + 2\log|\mathcal{Z}|$ as shown on the left-hand side of Eq. (5). Consequently, we can claim that multimodal contrastive learning is capable of identifying the latent variables $\mathbf{z}_x$ (symmetrically, $\mathbf{z}_t$) up to a linear transformation. Consequently, Corollary 4.2 suggests that models trained using multimodal contrastive learning on hypersphere, such as CLIP, can identify the latent variables $\mathbf{z}_x$ up to a linear transformation. This result highlights two key points: (1) strong support for the success of multimodal contrastive learning, as it can recover the true high-level latent coupled variables, and (2) the potential for disentanglement in models trained with multimodal contrastive loss, such as CLIP. Specifically, multimodal contrastive loss enables these models to learn features that correspond to a linear transformation of the true $\mathbf{z}_x$, e.g., $\mathbf{A}\mathbf{z}_x$. With this, linear ICA (Hyvärinen et al., 2001) can be applied to reduce the linear transformation $\mathbf{A}$ to a permutation matrix with scaling, facilitating component-wise recovery of $\mathbf{z}_x$. In our implementation, we employ the FastICA algorithm from (Hyvarinen, 1999) for this purpose. Note the geometry of the hypersphere in this context—specifically, the unit $M-1$-dimensional hypersphere—where the maximum number of independent dimensions is $M-1$. In other words, $\mathbf{z}_x$ can have at most $M-1$ independent components. Therefore, models trained with multimodal contrastive loss, such as CLIP, can achieve at most $M-1$ disentangled components via linear ICA.

## 4.2 IDENTIFIABILITY ANALYSIS ON CONVEX BODIES

The theoretical result above requires the latent coupled variables to be a hypersphere, this somehow limits the disentanglement ability of multimodal contrastive learning, due to the nature of the

geometric constraints on the hypersphere. In this section, we will show a similar result for convex bodies, *e.g.*, the hyperrectangle $[a_1, b_1] \times ... \times [a_M, b_M]$, which allows for independence among the latent coupled variables, offering a more flexible framework. On convex bodies, we parameterize the proposed latent partial causal generative models depicted in Figure 1 by the following:

$$p(\mathbf{z}_x) = |\mathcal{Z}_c|^{-1}, \quad p(\mathbf{z}_t|\mathbf{z}_x) = C_p(\mathbf{z}_x)^{-1}e^{-\delta(\mathbf{z}_t,\mathbf{z}_x)/\lambda}, \quad \mathbf{x} = \mathbf{g}_x(\mathbf{z}_x, \mathbf{m}_x), \quad \mathbf{t} = \mathbf{g}_t(\mathbf{z}_t, \mathbf{m}_t), \quad (7)$$

where $\delta$ is a distance metric induced by a norm. Diverging from the hypersphere space mentioned above, here we consider a convex body in $\mathbb{R}^M$, denoted as $\mathcal{Z}_c$. In this context, we assume that $p(\mathbf{z}_x)$ follows a uniform distribution, and the conditional distribution $p(\mathbf{z}_t|\mathbf{z}_x)$ follows an exponential distribution. Again, we do not enforce any further assumptions for $\mathbf{m}_x$ and $\mathbf{m}_t$. For $\mathbf{g}_x$ and $\mathbf{g}_t$, we assume them to be invertible mapping, ensuring information in latent space can be recovered. Given these assumptions on a convex body, we have the following result:

**Theorem 4.3.** *($\mathcal{L}$ converges to the symmetric cross-entropy) Under the assumptions defined in Eq. (7) for the proposed latent partial causal model, the necessary condition $\mathbf{f}_x \circ \mathbf{g}_x = \mathbf{f}_t \circ \mathbf{g}_t$, denoted as $\mathbf{h}$, for the optimal normalized multimodal contrastiveloss given by Eq. (2) leads to the following reduction of the loss itself:*

$$\lim_{N \to \infty} \mathcal{L} - 2\log N + 2\log|\mathcal{Z}_c| =$$
$$\underset{\mathbf{z}_x \sim p(\mathbf{z}_x)}{\mathbb{E}} \big[ H(p(\mathbf{z}_t|\mathbf{z}_x), q_{\mathbf{h}}(\mathbf{z}_t|\mathbf{z}_x)) \big] + \underset{\mathbf{z}_t \sim p(\mathbf{z}_t)}{\mathbb{E}} \big[ H(p(\mathbf{z}_x|\mathbf{z}_t), q_{\mathbf{h}}(\mathbf{z}_x|\mathbf{z}_t)) \big], \quad (8)$$

*where $H$ is the cross entropy, the conditional distributions $q_{\mathbf{h}}(\mathbf{z}_t|\mathbf{z}_x)$ and $q(\mathbf{z}_x|\mathbf{z}_t)$ are parameterized by the following:*

$$q_{\mathbf{h}}(\mathbf{z}_x|\mathbf{z}_t) = C_q(\mathbf{z}_t)e^{-\delta(\mathbf{h}(\mathbf{z}_x),\mathbf{h}(\mathbf{z}_t))/\tau}, q_{\mathbf{h}}(\mathbf{z}_t|\mathbf{z}_x) = C_q(\mathbf{z}_x)e^{-\delta(\mathbf{h}(\mathbf{z}_x),\mathbf{h}(\mathbf{z}_t))/\tau}, \quad (9)$$

with

$$C_q(\mathbf{z}_t) = \int e^{-\delta(\mathbf{h}(\mathbf{z}_x),\mathbf{h}(\mathbf{z}_t))/\tau}\mathrm{d}\mathbf{z}_x, C_q(\mathbf{z}_x) = \int e^{-\delta(\mathbf{h}(\mathbf{z}_x),\mathbf{h}(\mathbf{z}_t))/\tau}\mathrm{d}\mathbf{z}_t.$$

~~**A Bridge on Convex Bodies** Once again, regarding convex bodies, Theorem 4.3, first introduced in this work, is crucial for bridging multimodal contrastive loss with traditional contrastive loss by addressing various asymmetric challenges arising from differences between modalities. Leveraging this theorem, we can readily extend the findings of Theorem 5 in (Zimmermann et al., 2021) to a multimodal context, as follows: Specifically, Eq. (8) in theorem 4.3 can identify the latent variables $\mathbf{z}_x$ and $\mathbf{z}_t$ up to a permutation transformation, *i.e.*, the recovered latent variable $\hat{\mathbf{z}}_x$, obtained through the minimization of Eq. (5), is linearly related to the true $\mathbf{z}_x$ as follows: $\hat{\mathbf{z}}_x = \mathbf{P}\mathbf{z}_x + \mathbf{c}$, where $\mathbf{P}$ is an orthogonal matrix, $\mathbf{c}$ is a constant vector.~~

**Corollary 4.4.** *By leveraging Theorem 4.3, the minimization of Eq. (8) in theorem 4.3 identifies the latent variables $\mathbf{z}_x$ (symmetrically, $\mathbf{z}_t$) up to a permutation transformation, i.e., the recovered latent variable $\hat{\mathbf{z}}_x$, obtained through the minimization of Eq. (8), is related to the true $\mathbf{z}_x$ as follows: $\hat{\mathbf{z}}_x = \mathbf{P}\mathbf{z}_x + \mathbf{c}$, where $\mathbf{P}$ is an permutation matrix with scaling, $\mathbf{c}$ is a constant vector.*

For completeness, see details in Appendix A.4.2.

~~**Insights** Unlike hyperspheres, convex bodies allow for independent latent variables $\mathbf{z}_x$. Our Theorem 4.3 shows that minimizing Eq. (5) in a convex body setting can identify these variables up to permutation. Although CLIP was trained on a hypersphere, which differs from our convex body assumptions, it remains promising since it still aligns correct pairs and distances incorrect ones. The key to permutation identifiability is the isometry of the mapping $\mathbf{h}$ in Theorem 4.3. While a global isometry between a convex body and the entire hypersphere is not feasible, a local isometry between the convex body and a small hypersphere region is plausible. To leverage this, we can apply PCA to reduce redundant information learned by CLIP and then use FastICA to handle the orthogonal transformation introduced by PCA, extracting the final features.~~

**Connection with disentanglement** Unlike hyperspheres, convex bodies permit all components of the latent variables $\mathbf{z}_x$ to be independent. Corollary 4.4 demonstrates that minimizing Eq. (8) in a convex body setting can identify these variables up to a permutation. Although CLIP was trained on a hypersphere, differing from our convex body assumptions, it remains promising as it aligns

correct pairs while distancing incorrect ones. The crucial factor for permutation identifiability is the isometry of the mapping $\mathbf{h}$ in Corollary 4.4. While a global isometry between a convex body and an entire hypersphere is not feasible, a local isometry between a convex body and a small region of the hypersphere is plausible. To take advantage of this, PCA can be applied to reduce redundant information learned by CLIP, followed by FastICA to handle the orthogonal transformation introduced by PCA, ultimately extracting the final features.

## 5 EXPERIMENTS

**Experiments on Synthetic Data** In our initial experiments, we use synthetic data to verify our main identifiability results on hypersphere and convex bodies, and empirically demonstrate the robustness of these results when facing substantial violations of assumptions. For detailed information regarding the generation of synthetic data, please refer to Appendix A.9. We examine $p(\mathbf{z}_x)$ under conditions that align with our theoretical assumptions (using uniform distributions), and under conditions that deviate from our assumptions (using non-uniform distributions). Furthermore, we construct real pairs by sampling from the conditional distribution $p(\mathbf{z}_t|\mathbf{z}_x)$. This process is conducted in two distinct settings: one aligning with our assumptions about the conditional distribution and another that contravenes these assumptions. Beyond the hypersphere space, our experiments also encompass the bounded space and unbounded space. We conduct each experiment three times for each setting.

Table 1: Assessing identifiability up to linear (a) and permutation (b) transformations under varying assumptions. The first row corresponds to a setting that matches our assumptions in Theorem 4.1, while the others show results for violated assumptions. S: Space, Sp: Sphere, U: Uniform, v: vMF ($k = 1$), L: Laplace ($\lambda = 0.05$), N: Normal ($\delta = 0.05$), B: Box, Un: Unbounded, G: GenNorm($\beta = 3$).

(a) Assessing identifiability up to linear.

| | Generative process | | | Model | |
|---|---|---|---|---|---|
| S | $p(\mathbf{z}_x)$ | $p(\mathbf{z}_x|\mathbf{z}_t)$ | S | $q(\mathbf{z}_x|\mathbf{z}_t)$ | R2 |
| Sp | U | v | Sp | v | $99.5 \pm 0.1$ |
| Sp | U | L | Sp | v | $99.4 \pm 0.2$ |
| Sp | U | N | Sp | v | $98.7 \pm 0.3$ |
| B | U | N | Un | N | $90.5 \pm 0.2$ |
| B | U | L | Un | N | $92.2 \pm 0.3$ |
| B | U | L | Un | G | $99.1 \pm 0.4$ |
| B | U | N | Un | G | $91.2 \pm 0.3$ |
| Sp | N ($\delta = 1$) | L | Sp | v | $96.3 \pm 0.3$ |
| Sp | N ($\delta = 1$) | N | Sp | v | $95.9 \pm 0.2$ |
| Un | L ($\lambda = 1$) | N | Un | N | $88.5 \pm 0.3$ |
| Un | N ($\delta = 1$) | N | Un | N | $89.2 \pm 0.2$ |

(b) Assessing identifiability up to permutation.

| | Generative process | | | Model | |
|---|---|---|---|---|---|
| S | $p(\mathbf{z}_x)$ | $p(\mathbf{z}_x|\mathbf{z}_t)$ | S | $q(\mathbf{z}_x|\mathbf{z}_t)$ | MCC |
| B | U | L | B | L | $99.1 \pm 0.1$ |
| B | U | G | B | G | $97.2 \pm 0.3$ |
| B | U | N | B | N | $98.6 \pm 0.2$ |
| B | U | L | B | N | $99.1 \pm 0.1$ |
| B | U | G | B | L | $98.4 \pm 0.1$ |
| B | U | L | Un | L | $95.6 \pm 0.2$ |
| B | U | G | Un | G | $96.4 \pm 0.2$ |

To test for identifiability up to linear transformations formalized by ~~Theorem~~Corollary 4.2, we fit a linear regression model between the ground-truth $\mathbf{z}_x$ and recovered $\hat{\mathbf{z}}_x$ and report the coefficient of determination ($R^2$). Further, to test for identifiability up to permutations formalized by ~~Theorem~~Corollary 4.4, we employ the mean correlation coefficient (MCC). The first row in Table 1 (a) and the first two rows in Table 1 (b), corresponding to the setting where the assumptions are satisfied, verify the identifiability results on hypersphere and convex bodies, respectively. Our empirical investigations have yielded a critical insight: discrepancies in the assumptions concerning marginal and conditional distributions, as well as the nature of the spaces (hypersphere and convex body), do not significantly impact performance. This robustness is demonstrated by the results detailed in Table 1 (a) for the hypersphere space and Table 1 (b) for convex bodies. This observation is similar to reports from studies conducted in the context of single-model context (Zimmermann et al., 2021). This observation might be attributed to the fact that the loss function described in Eq. 2 predominantly relies on the computation of expectations, inherently allowing for a wide range of approximations. If we can approximate the expectation calculations consistently across various distributions and spaces, it is reasonable to expect that the identifiability results remain well within acceptable bounds.

**Disentangled representations for CelebA data** Informed by our identifiability results and the empirical evidence presented earlier, we have grounds to claim that the pre-trained CLIP model possesses disentanglement ability. To substantiate this, we first extract features from the pre-trained

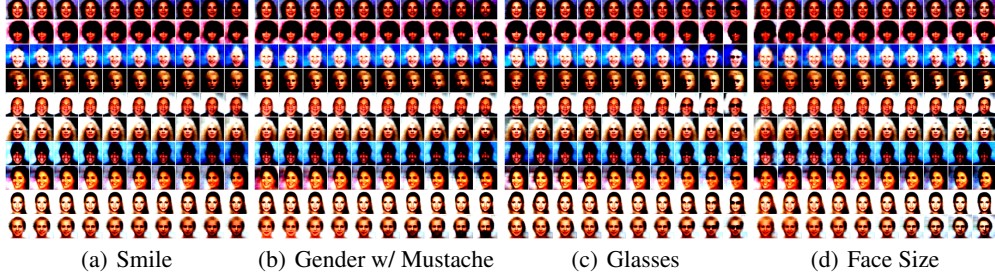

| (a) Smile | (b) Gender w/ Mustache | (c) Glasses | (d) Face Size |

Figure 3: Disentangled Representations learned by combining pre-train CLIP and FastICA. The proposed method obtains 16 disentangled representations, refer to Appendix A.6 for more results.

CLIP model and then apply FastICA to these features to achieve final representations. We expect these final representations to exhibit clear signs of disentanglement. To validate this, we proceed to train a decoder that reconstructs observational data using these extracted representations. We implement the above process on the CelebA face dataset (Liu et al., 2015), which has been explored by previous studies to learn disentangled representations (Kim and Mnih, 2018; Chen et al., 2018).

Figure 3 illustrates the effectiveness of our method through latent space traversals. Specifically, it visualizes changes in reconstructions as we traverse one dimension of the latent space at a time, showcasing 4 out of 16 attributes uncovered by our approach. Our method yields competitive results when compared with specialized techniques for learning disentangled representations, such as FactorVAE (Kim and Mnih, 2018) and $\beta$-TCVAE (Chen et al., 2018). As reported, FactorVAE identified 8 disentangled attributes and $\beta$-TCVAE reported 15, our method successfully discerns 16 distinct disentangled representations. Additional results are available in Appendix A.6. This achievement not only underscores the effectiveness of our method and validates our identifiability results, but also offers new perspectives into learning disentangled representations by CLIP.

**Few-shot learning and domain generalization** Broadly, the objective of disentangled representations is to learn features that lend themselves to be easily and robustly transferred to downstream tasks. This implies that disentangled representations should inherently possess a satisfactory capability for few-shot learning and demonstrate robustness against distribution shifts. Therefore, we focus on tasks involving few-shot learning and domain generalization, to validate our identifiability results and the efficacy of the proposed methods. To achieve this, we first obtain representations of a limited set of labeled samples. This is done either by utilizing the pre-trained CLIP model followed by FastICA (labeled as Linear Probe with FastICA, applied within the hypersphere space) or

Table 2: Quantitative results for 2-shot learning and domain generalization by different methods. ①: Linear Probe, ②: ① with FastICA, and ③: ① with PCA and FastICA.

| | | SOURCE | TARGET (IMAGENET-) | | | | |
|---|---|---|---|---|---|---|---|
| ENCODERS | METHODS | IMAGENET | V2 | SKETCH | R | A | AVG. |
| RN50 | ① | 31.95 | 26.48 | 8.41 | 20.74 | 7.44 | 15.77 |
| | ② | 34.06 | 28.74 | 8.37 | 21.72 | 10.15 | 17.25 |
| | ③ | 34.12 | 28.68 | 11.55 | 25.57 | 10.15 | 18.99 |
| RN101 | ① | 37.64 | 31.45 | 13.71 | 31.09 | 11.85 | 20.03 |
| | ② | 39.58 | 33.15 | 13.49 | 30.29 | 14.77 | 22.93 |
| | ③ | 39.86 | 33.58 | 17.93 | 35.48 | 14.20 | 25.29 |
| VIT32 | ① | 38.23 | 32.00 | 16.17 | 33.67 | 12.88 | 23.68 |
| | ② | 40.21 | 33.97 | 16.54 | 34.79 | 15.72 | 25.26 |
| | ③ | 39.34 | 33.44 | 19.02 | 36.98 | 14.69 | 26.03 |
| VIT16 | ① | 44.97 | 38.11 | 22.06 | 43.86 | 25.99 | 32.51 |
| | ② | 45.52 | 39.38 | 22.55 | 45.33 | 30.47 | 34.43 |
| | ③ | 46.57 | 40.66 | 26.67 | 49.69 | 31.48 | 37.13 |

by employing the pre-trained CLIP model in conjunction with PCA and FastICA (labeled as Linear Probe with PCA and FastICA, aligning with convex bodies). These extracted representations, along with their labels, are used to train a linear classifier. We train the proposed methods on ImageNet (Deng et al., 2009) with limited samples to evaluate their performance for few-shot learning, and also conduct evaluations on ImageNet-V2 (Recht et al., 2019), ImageNet-Sketch (Wang et al., 2019), ImageNet-R (Hendrycks et al., 2021a), and ImageNet-A (Hendrycks et al., 2021b) for demonstrating the robustness to distribution shift.

Table 2 present the performance metrics of the proposed methods in few-shot learning scenarios (as shown in the 'SOURCE' column) and in the context of distribution shift (as indicated in the 'TARGET' columns). An analysis of the data for the 'SOURCE' column in these tables reveals

that the proposed methods outperform the baseline approach of training a linear classifier with features directly obtained from pre-trained CLIP, known as Linear Probe. This superior performance underscores the enhanced adaptability of our proposed methods, particularly in tasks requiring rapid learning from limited data. Furthermore, observations from the 'TARGET' column demonstrate that our proposed methods also surpass the Linear Probe approach in terms of distribution shift, which affirms the robustness of our methods. See Appendix A.7 for more results.

**FastICA as a plug-and-play Tool for Few-Shot Learning**   Recent progress has demonstrated that even with a few of labeled training samples, CLIP's adaptability can be significantly enhanced. The key of leveraging pre-trained CLIP for few-shot learning lies in effectively harnessing the features extracted from CLIP on the limited labeled training samples. As previously mentioned, disentangled representations should inherently possess a satisfactory capability for few-shot learning.

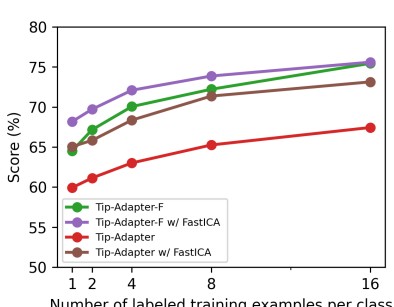

With this understanding, rather than directly utilizing CLIP's features, we can employ FastICA on the original CLIP's features to obtain disentangled ones, subsequently deploying them for few-shot learning tasks. This offers a plug-and-play integration of FastICA, resulting in performance improvements for existing methods. To verify this, following the experimental setup outlined in Zhang et al. (2022a), we incorporate FastICA after the original CLIP features, while preserving other components unchanged. Figure 4 show the results obtained by different few-shot CLIP adaptation methods over 11 datasets, including ImageNet (Deng et al., 2009), Caltech101 (Fei-Fei et al., 2004), FGVCAircraft (Maji et al., 2013), UCF101 (Soomro et al., 2012), EuroSAT (Helber et al., 2019), Flowers102 (Nilsback and Zisserman, 2008), StanfordCars (Krause et al., 2013), DTD (Cimpoi et al., 2014), Food101 (Bossard et al., 2014), OxfordPets (Parkhi et al., 2012), and, SUN397 (Xiao et al., 2010). A clear improvement over the original methods in Zhang et al. (2022a), including Tip-Adapter and Tip-Adapter-F, can be observed by incorporating FastICA. See Appendix A.8 for more details.

Figure 4: A comparison of accuracy (%) obtained by different few-shot CLIP adaptation methods over 11 datasets.

## 6    CONCLUSION

Instead of relying on traditional latent DAGs, we recognize that fully identifying latent causal models often requires various assumptions, which can be difficult to satisfy in real-world applications. This work explores latent partial causal models, where latent coupled variables—connected by an undirected edge—are used to model transferable knowledge across multimodal data. Our analysis reveals that the multimodal contrastive learning paradigm effectively identifies these latent coupled variables, which are critical for transferring knowledge between modalities. We also uncover a significant potential for disentanglement within multimodal contrastive learning, offering new insights and practical benefits for pre-trained models like CLIP. Our extensive experiments validate the robustness of these findings and demonstrate their practical implications for few-shot learning and domain generalization. Building on our findings, future work could explore applying our latent partial causal model to other multimodal learning paradigms, more importantly, unlocking new possibilities for the discovery of partial causal models.

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

# A APPENDIX

## A.1 THE PROOF OF THEOREM 3.1

**Theorem 2.1.** *(The asymptotics of $\mathcal{L}$) For fixed $\tau > 0$, as the sample size $N \to \infty$, the (normalized) multimodal contrastiveloss converges to*

$$
\lim_{N \to \infty} \mathcal{L} - 2 \log N = 2 \mathop{\mathbb{E}}_{(\mathbf{x},\mathbf{t}) \sim p(\mathbf{x},\mathbf{t})} \left[ d\big(\mathbf{f}_x(\mathbf{x}), \mathbf{f}_t(\mathbf{t})\big)/\tau \right] + \mathop{\mathbb{E}}_{\mathbf{x} \sim p(\mathbf{x})} \left[ \log \mathop{\mathbb{E}}_{\mathbf{t} \sim p(\mathbf{t})} \left[ e^{-d\big(\mathbf{f}_x(\mathbf{x}), \mathbf{f}_t(\mathbf{t})\big)/\tau} \right] \right] \quad (10)
$$
$$
+ \mathop{\mathbb{E}}_{\mathbf{t} \sim p(\mathbf{t})} \left[ \log \mathop{\mathbb{E}}_{\mathbf{x} \sim p(\mathbf{x})} \left[ e^{-d\big(\mathbf{f}_x(\mathbf{x}), \mathbf{f}_t(\mathbf{t})\big)/\tau} \right] \right].
$$

*Proof.* This proof is done by mainly depending on the Continuous Mapping Theorem and the law of large numbers.

$$
\lim_{N \to \infty} \mathcal{L} - 2 \log N = \lim_{N \to \infty} \bigg( -\frac{1}{N} \sum_{i=1}^{N} \log \frac{e^{-d\big(\mathbf{f}_x(\mathbf{x}_i), \mathbf{f}_t(\mathbf{t}_i)\big)/\tau}}{\sum_{j=1}^{N} e^{-d\big(\mathbf{f}_x(\mathbf{x}_i), \mathbf{f}_t(\mathbf{t}_j)\big)/\tau}}
$$
$$
-\frac{1}{N} \sum_{i=1}^{N} \log \frac{e^{-d\big(\mathbf{f}_x(\mathbf{x}_i), \mathbf{f}_t(\mathbf{t}_i)\big)/\tau}}{\sum_{j=1}^{N} e^{-d\big(\mathbf{f}_x(\mathbf{x}_j), \mathbf{f}_t(\mathbf{t}_i)\big)/\tau}} \bigg) - 2 \log N,
$$
$$
= \lim_{N \to \infty} \bigg( \frac{2}{N} \sum_{i=1}^{N} d\big(\mathbf{f}_x(\mathbf{x}_i), \mathbf{f}_t(\mathbf{t}_i)\big)/\tau + \frac{1}{N} \sum_{i=1}^{N} \log \sum_{j=1}^{N} e^{-d\big(\mathbf{f}_x(\mathbf{x}_i), \mathbf{f}_t(\mathbf{t}_j)\big)/\tau}
$$
$$
+ \frac{1}{N} \sum_{i=1}^{N} \log \sum_{j=1}^{N} e^{-d\big(\mathbf{f}_x(\mathbf{x}_j), \mathbf{f}_t(\mathbf{t}_i)\big)/\tau} \bigg) - 2 \log N
$$
$$
= \lim_{N \to \infty} \bigg( \frac{2}{N} \sum_{i=1}^{N} d\big(\mathbf{f}_x(\mathbf{x}_i), \mathbf{f}_t(\mathbf{t}_i)\big)/\tau + \frac{1}{N} \sum_{i=1}^{N} \log \frac{1}{N} \sum_{j=1}^{N} e^{-d\big(\mathbf{f}_x(\mathbf{x}_i), \mathbf{f}_t(\mathbf{t}_j)\big)/\tau}
$$
$$
+ \frac{1}{N} \sum_{i=1}^{N} \log \frac{1}{N} \sum_{j=1}^{N} e^{-d\big(\mathbf{f}_x(\mathbf{x}_j), \mathbf{f}_t(\mathbf{t}_i)\big)/\tau} + \frac{2}{N} \sum_{i=1}^{N} \log N \bigg) - 2 \log N
$$
$$
= 2 \mathop{\mathbb{E}}_{(\mathbf{x},\mathbf{t}) \sim p(\mathbf{x},\mathbf{t})} \left[ d\big(\mathbf{f}_x(\mathbf{x}), \mathbf{f}_t(\mathbf{t})\big)/\tau \right] + \mathop{\mathbb{E}}_{\mathbf{x} \sim p(\mathbf{x})} \left[ \log \mathop{\mathbb{E}}_{\mathbf{t} \sim p(\mathbf{t})} \left[ e^{-d\big(\mathbf{f}_x(\mathbf{x}), \mathbf{f}_t(\mathbf{t})\big)/\tau} \right] \right]
$$
$$
+ \mathop{\mathbb{E}}_{\mathbf{t} \sim p(\mathbf{t})} \left[ \log \mathop{\mathbb{E}}_{\mathbf{x} \sim p(\mathbf{x})} \left[ e^{-d\big(\mathbf{f}_x(\mathbf{x}), \mathbf{f}_t(\mathbf{t})\big)/\tau} \right] \right].
$$

$\square$

## A.2 RELATION WITH RECOVERING ALL INFORMATION

In this section, we proof

$$
\mathop{\mathbb{E}}_{\mathbf{x} \sim p(\mathbf{x})} \left[ \log \mathop{\mathbb{E}}_{\mathbf{t} \sim p(\mathbf{t})} \left[ e^{-d\big(\mathbf{f}_x(\mathbf{x}), \mathbf{f}_t(\mathbf{t})\big)/\tau} \right] \right] + \mathop{\mathbb{E}}_{\mathbf{t} \sim p(\mathbf{t})} \left[ \log \mathop{\mathbb{E}}_{\mathbf{x} \sim p(\mathbf{x})} \left[ e^{-d\big(\mathbf{f}_x(\mathbf{x}), \mathbf{f}_t(\mathbf{t})\big)/\tau} \right] \right]
$$
$$
\approx -H\big(p(\mathbf{f}_x(\mathbf{x})), p(\mathbf{f}_t(\mathbf{t}))\big) - H\big(p(\mathbf{f}_t(\mathbf{t})), p(\mathbf{f}_x(\mathbf{x}))\big).
$$

Considering the symmetry evident in both the left and right sides of the equation, let us focus our attention on the initial term on the left and its corresponding counterpart on the right.

$$
\mathop{\mathbb{E}}_{\mathbf{x} \sim p(\mathbf{x})} \left[ \log \mathop{\mathbb{E}}_{\mathbf{t} \sim p(\mathbf{t})} \left[ e^{-d\big(\mathbf{f}_x(\mathbf{x}), \mathbf{f}_t(\mathbf{t})\big)/\tau} \right] \right]
$$
$$
= \lim_{N \to \infty} \frac{1}{N} \sum_{i=1}^{N} \log \frac{1}{N} \sum_{j=1}^{N} e^{-d\big(\mathbf{f}_x(\mathbf{x}_i), \mathbf{f}_t(\mathbf{t}_j)\big)/\tau} \quad (11)
$$
$$
\approx \lim_{N \to \infty} \frac{1}{N} \sum_{i=1}^{N} \log p_{\text{KDE}}(\mathbf{f}_x(\mathbf{x}_i)) + \log Z_{\text{KDE}} \quad (12)
$$
$$
= -H(p(\mathbf{f}_x(\mathbf{x})), p(\mathbf{f}_t(\mathbf{t}))) + \log Z_{\text{KDE}}, \quad (13)
$$

Transitioning from Eq. (11) to Eq. (12), we employ kernel density estimation, wherein the choice of kernel is influenced by the distance metric used. For instance, on a hypersphere, a von Mises-Fisher kernel is suitable, whereas on convex bodies, a Laplace kernel aligns well with the L1 norm. In this context, $\log Z_{\text{KDE}}$ represents the normalization constant associated with the kernel. The inherent symmetry in this setup allows us to logically deduce the equation. Note that since here the bandwidth $\tau$ can be optimized in multimodel contrastive representation learning, if true distribution is the same as the chosen kernel, Eq. (12) is equal to Eq. (11), *i.e.*, $\approx$ in Eq. (12) can be $=$. Under certain conditions the kernel density estimation will converge to the real distbution, in that case $\approx$ in Eq. (12) can also be $=$.

## A.3 THE PROOF OF IDENTIFIABILITY ON HYPERSPHERE

### A.3.1 THE PROOF OF THEOREM 4.1

**Theorem 3.1.** *($\mathcal{L}$ converges to the symmetric cross-entropy) Under the assumptions defined in Eq. (4) for the proposed latent partial causal model, the necessary condition $\mathbf{f}_x \circ \mathbf{g}_x = \mathbf{f}_t \circ \mathbf{g}_t$, denoted as $\mathbf{h}$, for the optimal normalized multimodal contrasiveloss given by Eq. (2) leads to the following reduction of the loss itself:*

$$\lim_{N \to \infty} \mathcal{L} - 2 \log N = \mathbb{E}_{\mathbf{z}_x \sim p(\mathbf{z}_x)} \left[ H(p(\mathbf{z}_t|\mathbf{z}_x), q_{\mathbf{h}}(\mathbf{z}_t|\mathbf{z}_x)) \right] + \mathbb{E}_{\mathbf{z}_t \sim p(\mathbf{z}_t)} \left[ H(p(\mathbf{z}_x|\mathbf{z}_t), q_{\mathbf{h}}(\mathbf{z}_x|\mathbf{z}_t)) \right] \quad (14)$$

*where $H$ is the cross entropy, the conditional distributions $q_{\mathbf{h}}(\mathbf{z}_t|\mathbf{z}_x)$ and $q(\mathbf{z}_x|\mathbf{z}_t)$ are parameterized by the following:*

$$q_{\mathbf{h}}(\mathbf{z}_x|\mathbf{z}_t) = C_q(\mathbf{z}_t)^{-1} e^{(\mathbf{h}(\mathbf{z}_x)^T \mathbf{h}(\mathbf{z}_t)/\tau)}, \quad (15)$$

$$q_{\mathbf{h}}(\mathbf{z}_t|\mathbf{z}_x) = C_q(\mathbf{z}_x)^{-1} e^{(\mathbf{h}(\mathbf{z}_t)^T \mathbf{h}(\mathbf{z}_x)/\tau)}, \quad (16)$$

*with*

$$C_q(\mathbf{z}_t) = \int e^{(\mathbf{h}(\mathbf{z}_x)^T \mathbf{h}(\mathbf{z}_t)/\tau)} \mathrm{d}\mathbf{z}_x,$$

$$C_q(\mathbf{z}_x) = \int e^{(\mathbf{h}(\mathbf{z}_x)^T \mathbf{h}(\mathbf{z}_t)/\tau)} \mathrm{d}\mathbf{z}_t.$$

To proof Theorem 4.1, we first introduce the following Lemma.

**Lemma 4.1.** *Consider the unit hypersphere space, given uniform prior $p(\mathbf{z}_x)$, $p(\mathbf{z}_x) = |\mathcal{Z}|^{-1}$ where $\mathcal{Z} \subseteq \mathbb{R}^M$ denotes the space of $\mathbf{z}_x$, and conditional distribution $p(\mathbf{z}_t|\mathbf{z}_x) = C_p(k) \exp\left(k\mathbf{z}_x^T \mathbf{z}_t\right)$, $p(\mathbf{z}_t)$ follows a uniform distribution.*

*Proof.* By Bayesian theorem, $p(\mathbf{z}_t) = \int p(\mathbf{z}_t|\mathbf{z}_x)p(\mathbf{z}_x)\mathrm{d}\mathbf{z}_x = |\mathcal{Z}|^{-1} \int p(\mathbf{z}_t|\mathbf{z}_x)\mathrm{d}\mathbf{z}_x = |\mathcal{Z}|^{-1} C_p(k) \int \exp\left(k\mathbf{z}_x^T \mathbf{z}_t\right)\mathrm{d}\mathbf{z}_x$, then due to the unit hypersphere space, we have $\int \exp\left(k\mathbf{z}_x^T \mathbf{z}_t\right)\mathrm{d}\mathbf{z}_x = C_p(k)^{-1}$. As a result, we obtain $p(\mathbf{z}_t) = |\mathcal{Z}|^{-1}$. $\square$

**Lemma 5.1.** *The normalized multimodal contrastive loss in Eq. (2) has an optimal global solution of 0, which can be attained under the following conditions:*

- $\mathbf{h}_x(\mathbf{m}_x, \mathbf{z}_x) = \mathbf{h}_t(\mathbf{m}_t, \mathbf{z}_t)$ *almost surely, for pair $\left((\mathbf{m}_x, \mathbf{z}_x), (\mathbf{m}_t, \mathbf{z}_t)\right)$,  (C1),*

- $\mathbf{h}_x$ *and* $\mathbf{h}_t$ *map $(\mathbf{m}_x, \mathbf{z}_x)$ and $(\mathbf{m}_t, \mathbf{z}_t)$, respectively, to uniform variables on hypersphere,  (C2),*

*Proof.* First, it is well known that traditional contrastive loss in single modality has an optimal global solution of $\log N$ (Oord et al., 2018; Tian et al., 2020), as a result, the multimodal contrastive loss Eq. 1 has an optimal global solution of $2 \log N$. For completeness, let us focus on the first term in Eq. 1:

$$-\frac{1}{N} \sum_{i=1}^{N} \log \frac{e^{-d\left(\mathbf{f}_x(\mathbf{x}_i), \mathbf{f}_t(\mathbf{t}_i)\right)/\tau}}{\sum_{j=1}^{N} e^{-d\left(\mathbf{f}_x(\mathbf{x}_i), \mathbf{f}_t(\mathbf{t}_j)\right)/\tau}}, \quad (17)$$

Under optimal contrastive learning conditions, the distance for positive pairs satisfies: $e^{-d\left(\mathbf{f}_x(\mathbf{x}_i), \mathbf{f}_t(\mathbf{t}_i)\right)/\tau} = 1$, for negative pairs $(\mathbf{x}_i, \mathbf{x}_j)$ where $i \neq j$: $e^{-d\left(\mathbf{f}_x(\mathbf{x}_i), \mathbf{f}_t(\mathbf{t}_j)\right)/\tau} = \epsilon$, where $\epsilon$ is a small value. As a result, for each $i$, the denominator can be expressed as: $1 + (N-1)\epsilon$. Therefore, the first term in Eq. 1 reduces to : $-\frac{1}{N} \sum_{i=1}^{N} \log \frac{1}{1+(N-1)\epsilon}$. Clearly, when $N$ is large, the first term in Eq. 1 equals to $\log N$. Given that the second term is symmetric, we conclude that Eq. 1 has an optimal global solution of $2 \log N$. Therefore, Eq. 10 achieves a global optimal solution of 0. To reach the global minimum of 0, we observe that the first term in

Eq. 10 is minimized if and only if $\mathbf{h}_x(\mathbf{m}_x, \mathbf{z}_x) = \mathbf{h}_t(\mathbf{m}_t, \mathbf{z}_t)$ almost surely, for real pair $\big((\mathbf{m}_x, \mathbf{z}_x), (\mathbf{m}_t, \mathbf{z}_t)\big)$, (marked as (C1)). Thus, we obtain a minimum solution of 0 for the first term. Next, considering the remaining two terms in Eq. 10, as detailed in Appendix A.2, we see an equivalent expression: $-H(p(\mathbf{f}_x(\mathbf{x}), p(\mathbf{f}_t(\mathbf{t}))) - H(p(\mathbf{f}_x(\mathbf{x}), p(\mathbf{f}_t(\mathbf{t}))) + 2\log Z_{\text{KDE}}$. When both $\mathbf{h}_x$ and $\mathbf{h}_t$ map $(\mathbf{m}_x, \mathbf{z}_x)$ and $(\mathbf{m}_t, \mathbf{z}_t)$, respectively, to uniform variables on hypersphere (marked as (C2)), it reduces to $-2H(p(\mathbf{f}_x(\mathbf{x})) + 2\log Z_{\text{KDE}}$. Note that the entropy of a uniform distribution on the hypersphere $\mathbb{S}^{M-1}$ is $\log(\frac{2\pi^{M/2}}{\Gamma(M/2)})$, where $\Gamma$ is the gamma function. Together with the fact that the normalization constant of uniform distribution on hypersphere is $\log(\frac{\Gamma(M/2)}{2\pi^{M/2}})$ (i.e., $\log Z_{\text{KDE}}$), we arrive at the optimal solution of 0 for the last two terms. $\qquad\square$

**Proof sketch** The proof of Theorem 4.1 hinges on demonstrating the equality between the right-hand side of Eq. (14) and Eq. (10). Let us define $\mathbf{h}_x = \mathbf{f}_x \circ \mathbf{g}_x$ and $\mathbf{h}_t = \mathbf{f}_t \circ \mathbf{g}_t$. In Step I, using Lemma 5.1, we show that (1) $\mathbf{f}_x \circ \mathbf{g}_x = \mathbf{f}_t \circ \mathbf{g}_t$, and (2) they are independent of the modality-specific variables $\mathbf{m}_x$ and $\mathbf{m}_t$. In Step II, by defining $\mathbf{h} = \mathbf{f}_x \circ \mathbf{g}_x = \mathbf{f}_t \circ \mathbf{g}_t$ and applying both the generative model from Eq. (4) and the inference model from Eqs. (15)-(16), we establish the theorem.

**Step I** Consider C1 in Lemma 5.1, e.g., $\mathbf{h}_x(\mathbf{m}_x, \mathbf{z}_x) = \mathbf{h}_t(\mathbf{m}_t, \mathbf{z}_t)$ almost surely, for pair $\big((\mathbf{m}_x, \mathbf{z}_x), (\mathbf{m}_t, \mathbf{z}_t)\big)$, by differentiating it with respect to $\mathbf{m}_x$, we have:

$$\frac{\partial \mathbf{h}_x(\mathbf{m}_x, \mathbf{z}_x)}{\partial \mathbf{m}_x} = \frac{\partial \mathbf{h}_t(\mathbf{m}_t, \mathbf{z}_t)}{\partial \mathbf{m}_x} = 0, \tag{18}$$

, due to the independence between $\mathbf{m}_x$ and $(\mathbf{m}_t, \mathbf{z}_t)$. Similarly, by differentiating it with respect to $\mathbf{m}_t$, we have:

$$\frac{\partial \mathbf{h}_t(\mathbf{m}_t, \mathbf{z}_t)}{\partial \mathbf{m}_t} = \frac{\partial \mathbf{h}_x(\mathbf{m}_x, \mathbf{z}_x)}{\partial \mathbf{m}_t} = 0. \tag{19}$$

Based on Eqs. (18) and (19), we conclude that both $\mathbf{h}_x$ and $\mathbf{h}_t$ are independent of the modality-specific variables $\mathbf{m}_x$ and $\mathbf{m}_t$, respectively, i.e., $\mathbf{h}_x(\mathbf{m}_x, \mathbf{z}_x) = \mathbf{h}_x(\mathbf{z}_x)$ and $\mathbf{h}_t(\mathbf{m}_t, \mathbf{z}_t) = \mathbf{h}_t(\mathbf{z}_t)$. As a result, we have $\mathbf{h}_x(\mathbf{z}_x) = \mathbf{h}_t(\mathbf{z}_t)$, for all real pairs $(\mathbf{z}_x, \mathbf{z}_t)$ sampled from the conditional distribution $p(\mathbf{z}_t | \mathbf{z}_x)$ defined in Eq. (4). Note that this expression also holds true for $\mathbf{z}_t = \mathbf{z}_x$ (e.g., when $\mathbf{z}_t$ is sampled with the same value as $\mathbf{z}_x$), which implies $\mathbf{h}_x(\mathbf{z}_x) = \mathbf{h}_t(\mathbf{z}_x)$. As a result, we can obtain: $\mathbf{h}_x = \mathbf{h}_t$.

**Step II** According to the results above: $\mathbf{h}_x(\mathbf{m}_x, \mathbf{z}_x) = \mathbf{h}_x(\mathbf{z}_x)$, $\mathbf{h}_t(\mathbf{m}_t, \mathbf{z}_t) = \mathbf{h}_t(\mathbf{z}_x)$, and $\mathbf{h}_x = \mathbf{h}_t$ from Step I, by defining $\mathbf{h} \overset{\text{def}}{=} \mathbf{h}_x = \mathbf{h}_t$ , we can rewrite Eq. (10) as:

$$2 \underset{(\mathbf{z}_x, \mathbf{z}_t) \sim p(\mathbf{z}_x, \mathbf{z}_t)}{\mathbb{E}} \big[d\big(\mathbf{h}(\mathbf{z}_x), \mathbf{h}(\mathbf{z}_t)\big)/\tau\big] + \underset{\mathbf{z}_x \sim p(\mathbf{z}_x)}{\mathbb{E}} \Big[\log \underset{\mathbf{z}_t \sim p(\mathbf{z}_t)}{\mathbb{E}} \big[e^{-d\big(\mathbf{h}(\mathbf{z}_x), \mathbf{h}(\mathbf{z}_t)\big)/\tau}\big]\Big]$$

$$+ \underset{\mathbf{z}_t \sim p(\mathbf{z}_t)}{\mathbb{E}} \Big[\log \underset{\mathbf{z}_x \sim p(\mathbf{z}_x)}{\mathbb{E}} \big[e^{-d\big(\mathbf{h}(\mathbf{z}_x), \mathbf{h}(\mathbf{z}_t)\big)/\tau}\big]\Big]. \tag{20}$$

We then connect the right-hand side of Eq. (14) with Eq. (20). To this end, since the two terms in the right-hand side of Eq. (14) are symmetrical, we focus on one of the two terms for convenience, e.g., $\underset{\mathbf{z}_x \sim p(\mathbf{z}_x)}{\mathbb{E}} \big[H(p(\mathbf{z}_t | \mathbf{z}_x)), q_{\mathbf{h}}(\mathbf{z}_t | \mathbf{z}_x))\big]$. Based on Lemma 4.1, it can be shown that:

$$\underset{\mathbf{z}_x \sim p(\mathbf{z}_x)}{\mathbb{E}} \big[H(p(\mathbf{z}_t | \mathbf{z}_x)), q_{\mathbf{h}}(\mathbf{z}_t | \mathbf{z}_x))\big] \tag{21}$$

$$= \underset{\mathbf{z}_x \sim p(\mathbf{z}_x)}{\mathbb{E}} \big[ \underset{\mathbf{z}_t \sim p(\mathbf{z}_t | \mathbf{z}_x)}{\mathbb{E}} [-\log q_{\mathbf{h}}(\mathbf{z}_t | \mathbf{z}_x)]\big] \tag{22}$$

$$= \underset{(\mathbf{z}_x, \mathbf{z}_t) \sim p(\mathbf{z}_x, \mathbf{z}_t)}{\mathbb{E}} \big[ -\mathbf{h}(\mathbf{z}_x)^T \mathbf{h}(\mathbf{z}_t)/\tau + \log C_q(\mathbf{z}_x)\big] \tag{23}$$

$$= \underset{(\mathbf{z}_x, \mathbf{z}_t) \sim p(\mathbf{z}_x, \mathbf{z}_t)}{\mathbb{E}} \big[ -\mathbf{h}(\mathbf{z}_x)^T \mathbf{h}(\mathbf{z}_t)/\tau\big] + \underset{(\mathbf{z}_x) \sim p(\mathbf{z}_x)}{\mathbb{E}} [\log C_q(\mathbf{z}_x)] \tag{24}$$

$$= \underset{(\mathbf{z}_x, \mathbf{z}_t) \sim p(\mathbf{z}_x, \mathbf{z}_t)}{\mathbb{E}} \big[ -\mathbf{h}(\mathbf{z}_x)^T \mathbf{h}(\mathbf{z}_t)/\tau\big] + \underset{(\mathbf{z}_x) \sim p(\mathbf{z}_x)}{\mathbb{E}} [\log \int e^{(\mathbf{h}(\mathbf{z}_x)^T \mathbf{h}(\mathbf{z}_t)/\tau)} \mathrm{d}\mathbf{z}_x] \tag{25}$$

Since $p(\mathbf{z}_x) = |\mathcal{Z}|^{-1}$, and $p(\mathbf{z}_t) = |\mathcal{Z}|^{-1}$ by Lemma 4.1, Eq. (25) simplifies to:

$$= - \underset{(\mathbf{z}_x, \mathbf{z}_t) \sim p(\mathbf{z}_x, \mathbf{z}_t)}{\mathbb{E}} \big[(\mathbf{h}(\mathbf{z}_x)^T \mathbf{h}(\mathbf{z}_t))/\tau\big] + \underset{\mathbf{z}_x \sim p(\mathbf{z}_x)}{\mathbb{E}} \Big[\log \underset{\mathbf{z}_t \sim p(\mathbf{z}_t)}{\mathbb{E}} \big[e^{(\mathbf{h}(\mathbf{z}_x)^T \mathbf{h}(\mathbf{z}_t))/\tau}\big]\Big] + \log |\mathcal{Z}| \tag{26}$$

On hypersphere space with radius $r$, due to $\|\mathbf{h}(\mathbf{z}_x) - \mathbf{h}(\mathbf{z}_t)\| = 2r - 2\mathbf{h}(\mathbf{z}_x)^T \mathbf{h}(\mathbf{z}_t)$, Eq. 26 simplifies to:

$$= \underset{(\mathbf{z}_x, \mathbf{z}_t) \sim p(\mathbf{z}_x, \mathbf{z}_t)}{\mathbb{E}} \big[d\big(\mathbf{h}(\mathbf{z}_x), \mathbf{h}(\mathbf{z}_t)\big)/\tau\big] + \underset{\mathbf{z}_x \sim p(\mathbf{z}_x)}{\mathbb{E}} \Big[\log \underset{\mathbf{z}_t \sim p(\mathbf{z}_t)}{\mathbb{E}} \big[e^{-d\big(\mathbf{h}(\mathbf{z}_x)\mathbf{h}(\mathbf{z}_t)\big)/\tau}\big]\Big] \tag{27}$$

Similarly, for the second term in the right-hand side of Eq. (14), we can proof that:

$$\mathop{\mathbb{E}}_{(\mathbf{z}_t) \sim p(\mathbf{z}_t)} \big[ H(p(\mathbf{z}_x|\mathbf{z}_t)), q_\mathbf{h}(\mathbf{z}_x|\mathbf{z}_t)) \big] = \mathop{\mathbb{E}}_{(\mathbf{z}_x, \mathbf{z}_t) \sim p(\mathbf{z}_x, \mathbf{z}_t)} \big[ d\big(\mathbf{h}(\mathbf{z}_x), \mathbf{h}(\mathbf{z}_t)\big)/\tau \big]$$
$$+ \mathop{\mathbb{E}}_{\mathbf{z}_t \sim p(\mathbf{z}_t)} \Big[ \log \mathop{\mathbb{E}}_{\mathbf{z}_x \sim p(\mathbf{z}_x)} \big[ e^{-d\big(\mathbf{h}(\mathbf{z}_x), \mathbf{h}(\mathbf{z}_t)\big)/\tau} \big] \Big] + \log |\mathcal{Z}|. \quad (28)$$

By combining Eq. (27) and Eq. (28), we can conclude the proof.

### A.3.2 IDENTIFIABILITY RESULT ON HYPERSPHERE

Theorem 4.1 represents a adaptation of Theorem 1 from (Zimmermann et al., 2021) in the context of multi-modal setting. Specifically, within the confines of a single-modal framework, Theorem 4.1 is consistent with the findings presented in Theorem 1 in (Zimmermann et al., 2021). Consequently, this alignment allows us to employ Propositions 1 and 2 from (Zimmermann et al., 2021) to demonstrate that the global minimization of the objective outlined in Eq. (5), as specified in Theorem 4.1, identifies the latent variables $\mathbf{z}_x$, as well as $\mathbf{z}_x$, up to linear transformations. For completeness, a brief proof is provided herein, with comprehensive details available in the original work. Clearly, the global minimum of the cross-entropy between two distributions is reached if they match by value and have the same support. Therefore, for the optimal solution of the objective loss Eq. (14) in Theorem 4.1, we have:

$$p(\mathbf{z}_t|\mathbf{z}_x) = q_\mathbf{h}(\mathbf{z}_t|\mathbf{z}_x), \quad (29)$$

This expression also holds true for $\mathbf{z}_t = \mathbf{z}_x$; additionally using that $\mathbf{h}$ maps from a unit hypersphere to one with radius $\sqrt{\tau k}$, we have:

$$C_p^{-1} e^{(k\mathbf{z}_x^T \mathbf{z}_x)} = C_q(\mathbf{z}_x)^{-1} e^{(\mathbf{h}(\mathbf{z}_x)^T \mathbf{h}(\mathbf{z}_x)/\tau)},$$
$$\Leftrightarrow C_p = C_q(\mathbf{z}_x) \quad (30)$$

As the normalization constants are identical we get for all $\mathbf{z}_x, \mathbf{z}_t$,

$$k\mathbf{z}_x^T \mathbf{z}_t = \mathbf{h}(\mathbf{z}_x)^T \mathbf{h}(\mathbf{z}_t)/\tau, \quad (31)$$

here we can see that $\mathbf{h}$ maintains the dot product, which implies that $\mathbf{h}$ must be an orthogonal linear transformation by using Proposition 2 in (Zimmermann et al., 2021). As a result, Theorem 4.1 is capable of identifying the latent variables $\mathbf{z}_x$ and $\mathbf{z}_t$ up to an orthogonal linear transformation, *i.e.*, the recovered latent variable $\hat{\mathbf{z}}_x$, obtained through the minimization of Eq. (5), is linearly related to the true $\mathbf{z}_x$ as follows: $\hat{\mathbf{z}}_x = \mathbf{A}\mathbf{z}_x + \mathbf{c}$, where $\mathbf{A}$ represents an orthogonal matrix, and $\mathbf{c}$ is a constant vector.

## A.4 THE PROOF OF IDENTIFIABILITY ON CONVEX BODIES

### A.4.1 THE PROOF OF THEOREM 4.3

**Theorem 3.2.** (*$\mathcal{L}$ converges to the symmetric cross-entropy*) *Under the assumptions defined in Eqs. (7)-(7) for the proposed latent partial causal model, the necessary condition $\mathbf{f}_x \circ \mathbf{g}_x = \mathbf{f}_t \circ \mathbf{g}_t$, denoted as $\mathbf{h}$, for the optimal normalized multimodal contrastiveloss given by Eq. (2) leads to the following reduction of the loss itself:*

$$\lim_{N \to \infty} \mathcal{L} - 2\log N = \mathop{\mathbb{E}}_{\mathbf{z}_x \sim p(\mathbf{z}_x)} \big[ H(p(\mathbf{z}_t|\mathbf{z}_x)), q_\mathbf{h}(\mathbf{z}_t|\mathbf{z}_x)) \big] + \mathop{\mathbb{E}}_{(\mathbf{z}_t) \sim p(\mathbf{z}_t)} \big[ H(p(\mathbf{z}_x|\mathbf{z}_t)), q_\mathbf{h}(\mathbf{z}_x|\mathbf{z}_t)) \big] \quad (32)$$

*where $H$ is the cross entropy, the conditional distributions $q_\mathbf{h}(\mathbf{z}_t|\mathbf{z}_x)$ and $q(\mathbf{z}_x|\mathbf{z}_t)$ are parameterized by the following:*

$$q_\mathbf{h}(\mathbf{z}_x|\mathbf{z}_t) = C_q(\mathbf{z}_t)^{-1} e^{-\delta(\mathbf{h}(\mathbf{z}_x), \mathbf{h}(\mathbf{z}_t))/\tau}, \quad (33)$$
$$q_\mathbf{h}(\mathbf{z}_t|\mathbf{z}_x) = C_q(\mathbf{z}_x)^{-1} e^{-\delta(\mathbf{h}(\mathbf{z}_x), \mathbf{h}(\mathbf{z}_t))/\tau}, \quad (34)$$

*with*

$$C_q(\mathbf{z}_t) = \int e^{-\delta(\mathbf{h}(\mathbf{z}_x), \mathbf{h}(\mathbf{z}_t))/\tau} \mathrm{d}\mathbf{z}_x,$$
$$C_q(\mathbf{z}_x) = \int e^{-\delta(\mathbf{h}(\mathbf{z}_x), \mathbf{h}(\mathbf{z}_t))/\tau} \mathrm{d}\mathbf{z}_t.$$

Similar to the proof A.3.1, we first introduce the following Lemma.

**Lemma 4.2.** *For random variables $\mathbf{z}_x \in \mathcal{Z}_c$ and $\mathbf{z}_t = \mathcal{Z}_c$, assume that $p(\mathbf{z}_x) = 1/|\mathcal{Z}_c|$ if $\mathbf{z}_x \in \mathcal{Z}_c$ and 0 otherwise, and assume that conditional distribution $p(\mathbf{z}_t|\mathbf{z}_x) = C(\mathbf{z}_x) \exp\left(-\delta(\mathbf{z}_x, \mathbf{z}_t)/\lambda\right)$, where $\delta$ is a symmetric metric induced by a norm, then $p(\mathbf{z}_t)$ converges to uniform distribution on $\mathcal{Z}_c$ as $\lambda \to 0_+$.*

*Proof.* The proof can be done by the fact that as $\lambda \to 0$, the condition distribution $p(\mathbf{z}_t|\mathbf{z}_x)$ converges to a delta distribution, resulting that $p(\mathbf{z}_t) = p(\mathbf{z}_x)$. More specifically, as we will let $\lambda \to 0$ in the procedure, it is notable that the normalize $C(\mathbf{z}_x)$ actually depend on $\lambda$ and should be write as $C(\mathbf{z}_x, \lambda)$ in a more formal way. With simple integration trick, it would be straightforward to show that $C(\mathbf{z}_x, \lambda)$ can be decomposed as $C(\mathbf{z}_x, \lambda) = \frac{1}{\lambda} C'(\mathbf{z}_x)$.

By definition we have

$$
\begin{aligned}
p(\mathbf{z}_t) &= \int_{\mathbf{z}_x \in \mathcal{Z}_c} p(\mathbf{z}_x) p(\mathbf{z}_t|\mathbf{z}_x) \mathrm{d}\mathbf{z}_x \\
&= \int_{\mathbf{z}_x \in \mathcal{Z}_c} p(\mathbf{z}_x) \frac{1}{\lambda} C'(\mathbf{z}_x) \exp\left(-\delta(\mathbf{z}_x, \mathbf{z}_t)/\lambda\right) \mathrm{d}\mathbf{z}_x \\
&= \lim_{N \to +\infty} \sum_{i=1}^{N} \frac{1}{\lambda} C'(\mathbf{z}_{x_i}) \exp\left(-\delta(\mathbf{z}_{x_i}, \mathbf{z}_t)/\lambda\right), \forall i, \ \mathbf{z}_{x_i} \sim p(\mathbf{z}_x)
\end{aligned}
\tag{35}
$$

then obviously we have that

$$
\begin{aligned}
\lim_{\lambda \to 0_+} p(\mathbf{z}_t) &= \lim_{\lambda \to 0_+} \lim_{N \to +\infty} \sum_{i=1}^{N} \frac{1}{\lambda} C'(\mathbf{z}_{x_i}) \exp\left(-\delta(\mathbf{z}_{x_i}, \mathbf{z}_t)/\lambda\right) \\
&= \lim_{\lambda \to 0_+} \lim_{N \to +\infty} \sum_{i=1}^{N} \frac{1}{\lambda} C' \exp\left(-\delta(\mathbf{z}_{x_i}, \mathbf{z}_t)/\lambda\right),
\end{aligned}
\tag{36}
$$

where $C' = \int_{-\infty}^{\infty} \exp\left(-\delta(\mathbf{0}, \mathbf{z}_t)\right) \mathrm{d}\mathbf{z}_t$. It is obvious that (36) can be viewed as a Kernel Density Estimation over samples $\mathbf{z}_{x_i} \sim p(\mathbf{z}_x)$, and obviously $\lim_{\tau \to 0_+} p(\mathbf{z}_t)$ will converge to $p(\mathbf{z}_x)$ (which is uniform distribution) under quite mild condition (for details of the convergence we refer to (Jiang, 2017)). □

**Proof sketch** Similar to hypersphere, the proof of Theorem 4.3 can be done by demonstrating that the right-hand side of Eq. (32) is equal to the right-hand side of Eq. (10) on convex bodies. To achieve this, using Lemma 5.1, we show that $\mathbf{f}_x \circ \mathbf{g}_x = \mathbf{f}_t \circ \mathbf{g}_t$, and they are independent of the modality-specific variables $\mathbf{m}_x$ and $\mathbf{m}_t$, respectively. Finally, by defining $\mathbf{h} = \mathbf{f}_x \circ \mathbf{g}_x = \mathbf{f}_t \circ \mathbf{g}_t$, and using the inference model (33) and (34), we obtain our result.

**Step I** On convex bodies, and define $\mathbf{h}_x = \mathbf{f}_x \circ \mathbf{g}_x$ and $\mathbf{h}_t = \mathbf{f}_t \circ \mathbf{g}_t$. Consider C1 in Lemma 5.1, e.g., $\mathbf{h}_x(\mathbf{m}_x, \mathbf{z}_x) = \mathbf{h}_t(\mathbf{m}_t, \mathbf{z}_t)$ almost surely, for pair $\left((\mathbf{m}_x, \mathbf{z}_x), (\mathbf{m}_t, \mathbf{z}_t)\right)$. Similar to Step I in Appendix A.3.1, by differentiating it with respect to $\mathbf{m}_x$ and $\mathbf{m}_t$, respectively, we can conclude that both $\mathbf{h}_x$ and $\mathbf{h}_t$ are independent of the modality-specific variables $\mathbf{m}_x$ and $\mathbf{m}_t$, respectively, i.e., $\mathbf{h}_x(\mathbf{m}_x, \mathbf{z}_x) = \mathbf{h}_x(\mathbf{z}_x)$ and $\mathbf{h}_t(\mathbf{m}_t, \mathbf{z}_t) = \mathbf{h}_t(\mathbf{z}_t)$. Further, since $\mathbf{h}_x(\mathbf{z}_x) = \mathbf{h}_t(\mathbf{z}_t)$ hold, for all real pairs $(\mathbf{z}_x, \mathbf{z}_t)$ sampled from the conditional distribution $p(\mathbf{z}_t|\mathbf{z}_x)$ defined in Eq. (7), this expression also holds true for $\mathbf{z}_t = \mathbf{z}_x$, which implies $\mathbf{h}_x(\mathbf{z}_x) = \mathbf{h}_t(\mathbf{z}_x)$. As a result, we can obtain: $\mathbf{h}_x = \mathbf{h}_t$.

**Step II** According to the results above: $\mathbf{h}_x(\mathbf{m}_x, \mathbf{z}_x) = \mathbf{h}_x(\mathbf{z}_x)$, $\mathbf{h}_t(\mathbf{m}_t, \mathbf{z}_t) = \mathbf{h}_t(\mathbf{z}_t)$, and $\mathbf{h}_x = \mathbf{h}_t$, by defining $\mathbf{h} \overset{\text{def}}{=} \mathbf{f}_x \circ \mathbf{g}_x = \mathbf{f}_t \circ \mathbf{g}_t$ , we can rewrite Eq. (10) as:

$$
\begin{aligned}
2 \mathop{\mathbb{E}}_{(\mathbf{z}_x, \mathbf{z}_t) \sim p(\mathbf{z}_x, \mathbf{z}_t)} &\left[d\big(\mathbf{h}(\mathbf{z}_x), \mathbf{h}(\mathbf{z}_t)\big)/\tau\right] + \mathop{\mathbb{E}}_{\mathbf{z}_x \sim p(\mathbf{z}_x)} \left[\log \mathop{\mathbb{E}}_{\mathbf{z}_t \sim p(\mathbf{z}_t)} \left[e^{-d\big(\mathbf{h}(\mathbf{z}_x), \mathbf{h}(\mathbf{z}_t)\big)/\tau}\right]\right] \\
&+ \mathop{\mathbb{E}}_{\mathbf{z}_t \sim p(\mathbf{z}_t)} \left[\log \mathop{\mathbb{E}}_{\mathbf{z}_x \sim p(\mathbf{z}_x)} \left[e^{-d\big(\mathbf{h}(\mathbf{z}_x), \mathbf{h}(\mathbf{z}_t)\big)/\tau}\right]\right].
\end{aligned}
\tag{37}
$$

We then connect the right-hand side of Eq. (32) with Eq. (37). To this end, since the two terms in the right-hand side of Eq. (32) are symmetrical, we focus on one of the two terms for convenience, e.g.,

$\underset{\mathbf{z}_x \sim p(\mathbf{z}_x)}{\mathbb{E}} \big[ H(p(\mathbf{z}_t|\mathbf{z}_x)), q_{\mathbf{h}}(\mathbf{z}_t|\mathbf{z}_x)) \big]$. It can be shown that:

$$\underset{\mathbf{z}_x \sim p(\mathbf{z}_x)}{\mathbb{E}} \big[ H(p(\mathbf{z}_t|\mathbf{z}_x)), q_{\mathbf{h}}(\mathbf{z}_t|\mathbf{z}_x)) \big] \tag{38}$$

$$= \underset{\mathbf{z}_x \sim p(\mathbf{z}_x)}{\mathbb{E}} \Big[ \underset{\mathbf{z}_t \sim p(\mathbf{z}_t|\mathbf{z}_x)}{\mathbb{E}} [-\log q_{\mathbf{h}}(\mathbf{z}_t|\mathbf{z}_x)] \Big] \tag{39}$$

$$= \underset{(\mathbf{z}_x, \mathbf{z}_t) \sim p(\mathbf{z}_x, \mathbf{z}_t)}{\mathbb{E}} \big[ \delta(\mathbf{h}(\mathbf{z}_x), \mathbf{h}(\mathbf{z}_t))/\tau + \log C_q(\mathbf{z}_x) \big] \tag{40}$$

$$= \underset{(\mathbf{z}_x, \mathbf{z}_t) \sim p(\mathbf{z}_x, \mathbf{z}_t)}{\mathbb{E}} \big[ \delta(\mathbf{h}(\mathbf{z}_x), \mathbf{h}(\mathbf{z}_t))/\tau \big] + \underset{(\mathbf{z}_x) \sim p(\mathbf{z}_x)}{\mathbb{E}} [\log C_q(\mathbf{z}_x)] \tag{41}$$

$$= \underset{(\mathbf{z}_x, \mathbf{z}_t) \sim p(\mathbf{z}_x, \mathbf{z}_t)}{\mathbb{E}} \big[ \delta(\mathbf{h}(\mathbf{z}_x), \mathbf{h}(\mathbf{z}_t))/\tau \big] + \underset{(\mathbf{z}_x) \sim p(\mathbf{z}_x)}{\mathbb{E}} [\log \int e^{(-\delta(\mathbf{h}(\mathbf{z}_x), \mathbf{h}(\mathbf{z}_t))/\tau)} \mathrm{d}\mathbf{z}_x] \tag{42}$$

Since $p(\mathbf{z}_x) = |\mathcal{Z}|^{-1}$, and $p(\mathbf{z}_t) = |\mathcal{Z}|^{-1}$ by Lemma 4.2, Eq. (42) is equal to:

$$= \underset{(\mathbf{z}_x, \mathbf{z}_t) \sim p(\mathbf{z}_x, \mathbf{z}_t)}{\mathbb{E}} \big[ \delta(\mathbf{h}(\mathbf{z}_x), \mathbf{h}(\mathbf{z}_t))/\tau \big] + \underset{\mathbf{z}_x \sim p(\mathbf{z}_x)}{\mathbb{E}} \Big[ \log \underset{\mathbf{z}_t \sim p(\mathbf{z}_t)}{\mathbb{E}} \big[ e^{-\delta\big(\mathbf{h}(\mathbf{z}_x), \mathbf{h}(\mathbf{z}_t)\big)/\tau} \big] \Big] + \log |\mathcal{Z}_c| \tag{43}$$

Similarly, for the second term in the right-hand side of Eq. (32), we can proof that:

$$\underset{(\mathbf{z}_t) \sim p(\mathbf{z}_t)}{\mathbb{E}} \big[ H(p(\mathbf{z}_x|\mathbf{z}_t)), q_{\mathbf{h}}(\mathbf{z}_x|\mathbf{z}_t)) \big] = \underset{(\mathbf{z}_x, \mathbf{z}_t) \sim p(\mathbf{z}_x, \mathbf{z}_t)}{\mathbb{E}} \big[ \delta\big(\mathbf{h}(\mathbf{z}_x), \mathbf{h}(\mathbf{z}_t)\big)/\tau \big] \tag{44}$$

$$+ \underset{\mathbf{z}_t \sim p(\mathbf{z}_t)}{\mathbb{E}} \Big[ \log \underset{\mathbf{z}_x \sim p(\mathbf{z}_x)}{\mathbb{E}} \big[ e^{-\delta\big(\mathbf{h}(\mathbf{z}_x), \mathbf{h}(\mathbf{z}_t)\big)/\tau} \big] \Big] + \log |\mathcal{Z}_c|. \tag{45}$$

By combining Eq. (43) and Eq. (45), we can conclude the proof.

### A.4.2 IDENTIFIABILITY RESULT ON CONVEX BODIES

Theorem 4.3 represents a symmetrical adaptation of Theorem 3 from (Zimmermann et al., 2021). This alignment allows us to employ Propositions 4, Lemma 1 and Lemma A from (Zimmermann et al., 2021) to demonstrate that the global minimization of the objective outlined in Eq. (32), as specified in Theorem 4.3, identifies the latent variables $\mathbf{z}_x$, as well as $\mathbf{z}_x$, up to linear transformations. For completeness, a brief proof is provided herein, with comprehensive details available in the original work. Clearly, the global minimum of the cross-entropy between two distributions is reached if they match by value and have the same support. Therefore, for the optimal solution of the objective loss Eq. (10) in Theorem 4.3, we have:

$$p(\mathbf{z}_t|\mathbf{z}_x) = q_{\mathbf{h}}(\mathbf{z}_t|\mathbf{z}_x), \tag{46}$$

This expression also holds true for $\mathbf{z}_t = \mathbf{z}_x$, we have:

$$C_p(\mathbf{z}_x)^{-1} e^{-\delta(\mathbf{z}_x, \mathbf{z}_x)/\lambda} = C_q(\mathbf{z}_x)^{-1} e^{-\delta(\mathbf{h}(\mathbf{z}_x), \mathbf{h}(\mathbf{z}_x))/\tau},$$
$$\Leftrightarrow C_p(\mathbf{z}_x) = C_q(\mathbf{z}_x) \tag{47}$$

As the normalization constants are identical we get for all $\mathbf{z}_x, \mathbf{z}_t$,

$$\delta(\mathbf{z}_x, \mathbf{z}_t) = \lambda \delta(\mathbf{h}(\mathbf{z}_x), \mathbf{h}(\mathbf{z}_t))/\tau. \tag{48}$$

Then, by limiting $\delta$ be an $L^\alpha$ metric for $\alpha \geq 1$, $\alpha \neq 2$ or the $\alpha$-th power of such an $L^\alpha$ metric, using the Theorem 5 in (Zimmermann et al., 2021), Theorem 4.1 can identify the latent variables $\mathbf{z}_x$ and $\mathbf{z}_t$ up to an permutation transformation, *i.e.*, the recovered latent variable $\hat{\mathbf{z}}_x$, obtained through the minimization of Eq. (8), is related to the true $\mathbf{z}_x$ as follows: $\hat{\mathbf{z}}_x = \mathbf{P}\mathbf{z}_x + \mathbf{c}$, where $\mathbf{P}$ represents an permutation matrix with scaling, and $\mathbf{c}$ is a constant vector.

### A.5 DIFFERENCES WITH PREVIOUS WORKS IN IDENTIFIABILITY RESULT

The differences in formulating the causal generative model naturally results in differences in identifiability results. The identifiability results obtained in this work diverge from those found in previous works (Daunhawer et al., 2023; Yao et al., 2023), both in terms of breadth and depth of identifiability, due to the introduction of the undirected edge between $\mathbf{z}_x$ and $\mathbf{z}_t$. a) Breadth of Identifiability: Unlike earlier works that often achieve only partial identifiability of latent coupled variables $\mathbf{z}_x$ and $\mathbf{z}_t$, *e.g.*, latent content variables but not latent style variables (Daunhawer et al., 2023; Yao et al., 2023), our model extends this scope to ensure complete identifiability of latent coupled variables $\mathbf{z}_x$ and $\mathbf{z}_t$. b) Depth of Identifiability: In terms of depth, this work identifies latent coupled variables $\mathbf{z}_x$ and $\mathbf{z}_t$ up to linear or permutation transformations. This level of precision offers an enhancement over the block identifiability result in previous studies (Daunhawer et al., 2023; Yao et al., 2023). The differences above in both breadth and depth of identifiability results enable us, for the first time, to unveil the disentanglement capabilities of multimodal contrastive representation learning.

## A.6    MORE RESULTS ON CELEBA

(a) Smile

(b) Brightness

(c) Hair color

(d) Eye shadow

(e) Gender w/ Mustache

(f) Glasses

Figure 5: Disentangled Representations learned by combining pre-train CLIP and FastICA.

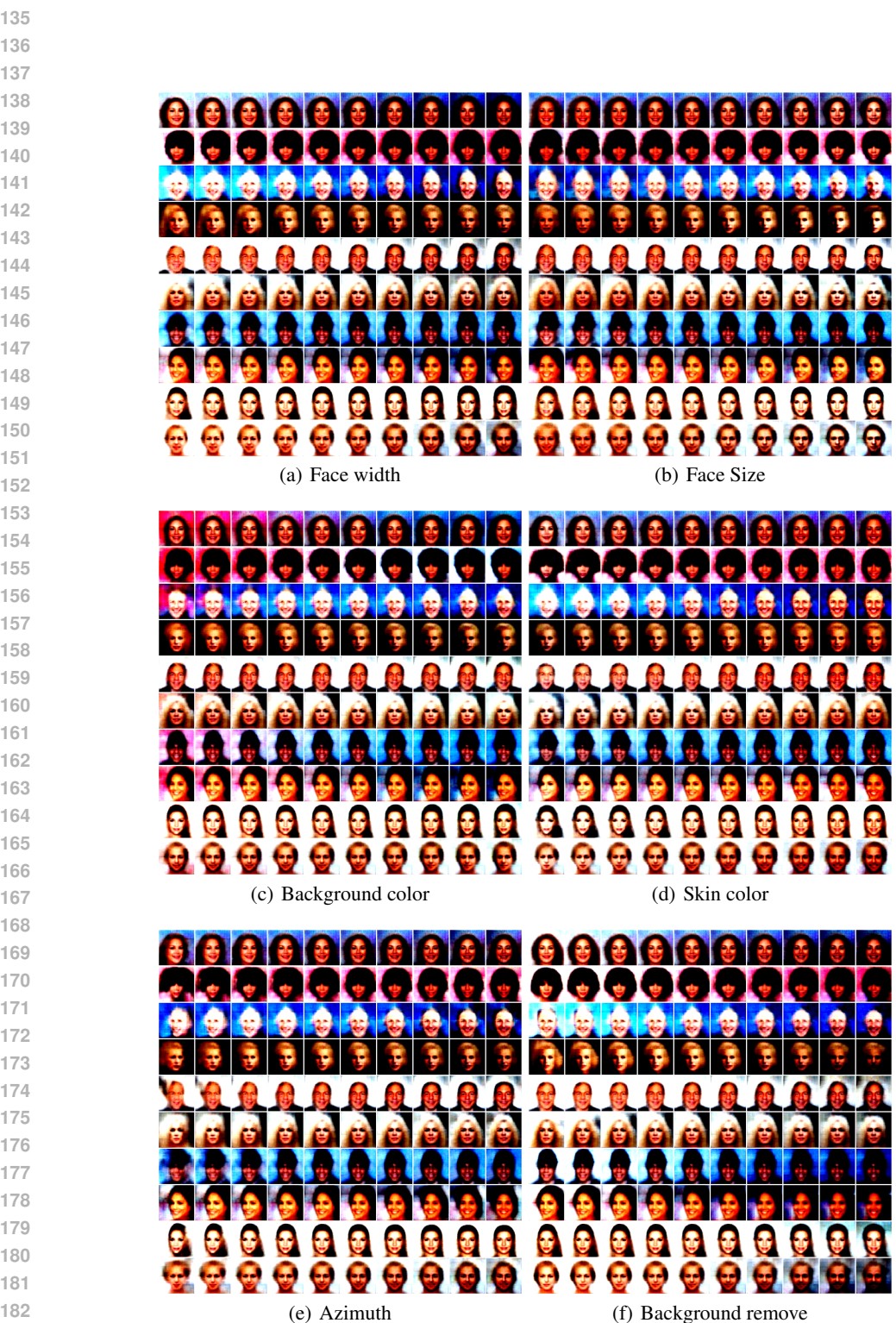

(a) Face width

(b) Face Size

(c) Background color

(d) Skin color

(e) Azimuth

(f) Background remove

Figure 6: Disentangled Representations learned by combining pre-train CLIP and FastICA.

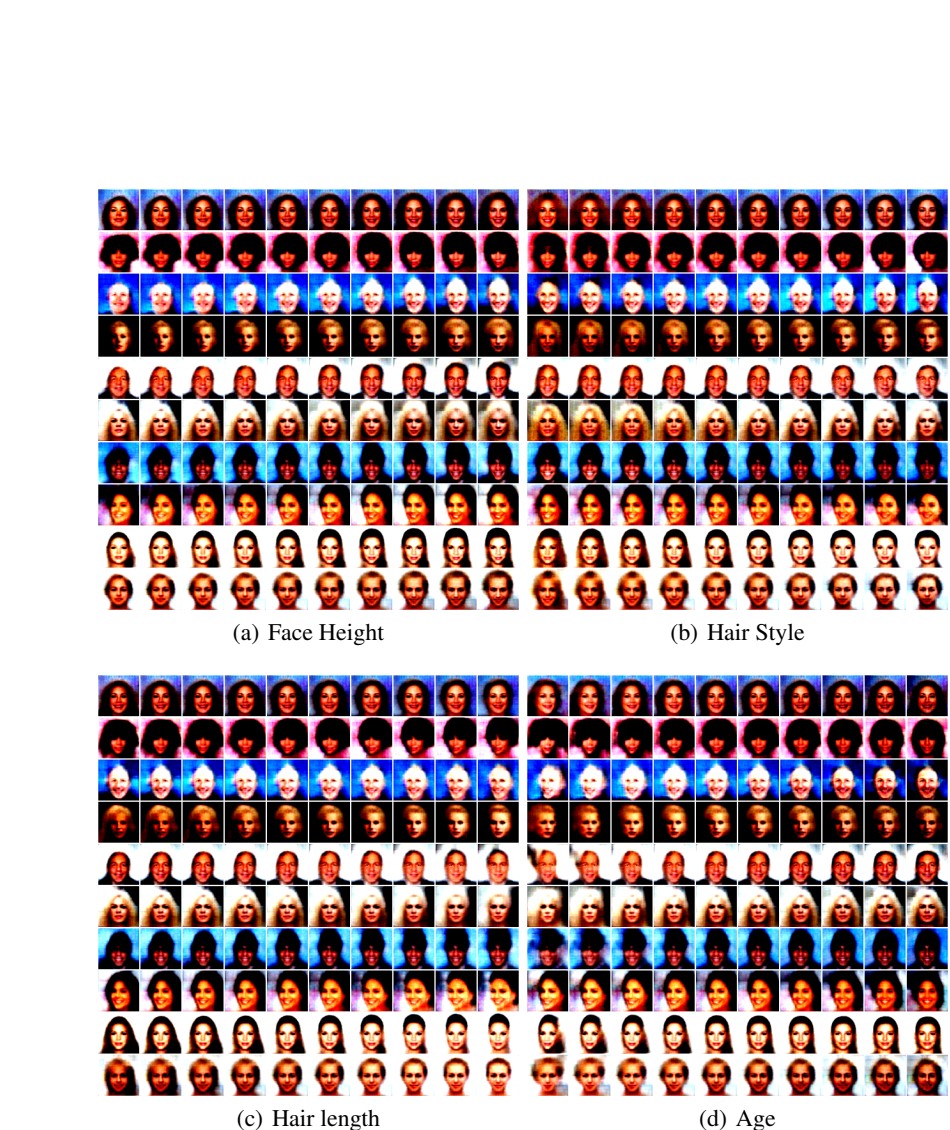

(a) Face Height                    (b) Hair Style

(c) Hair length                    (d) Age

Figure 7: Disentangled Representations learned by combining pre-train CLIP and FastICA.

## A.7 MORE RESULTS ON IMAGENET-TYPE DATA

Table 3: Quantitative results for 16-shot transfer learning and domain generalization by different methods. Lin. P. (Linear Probe).

| ENCODERS | METHODS | SOURCE IMAGENET | V2 | SKETCH | R | A | AVG. |
|---|---|---|---|---|---|---|---|
| RN50 | LIN. P. | 55.36 | 45.45 | 18.22 | 34.09 | 12.52 | 27.77 |
| | LIN. P. W/ FASTICA | 57.82 | 47.78 | 19.77 | 38.05 | 13.15 | 29.69 |
| | LIN. P. W/ PCA AND FASTICA | 57.37 | 47.67 | 20.39 | 38.76 | 12.89 | 29.93 |
| RN101 | LIN. P. | 60.98 | 50.36 | 25.80 | 46.61 | 18.64 | 35.35 |
| | LIN. P. W/ FASTICA | 61.86 | 51.85 | 27.29 | 49.29 | 19.89 | 37.08 |
| | LIN. P. W/ PCA AND FASTICA | 61.58 | 51.44 | 28.86 | 50.32 | 19.97 | 37.64 |
| VIT32 | LIN. P. | 60.76 | 50.92 | 28.81 | 49.18 | 19.72 | 37.15 |
| | LIN. P. W/ FASTICA | 61.94 | 51.95 | 30.30 | 51.82 | 20.81 | 38.72 |
| | LIN. P. W/ PCA AND FASTICA | 62.00 | 52.39 | 30.39 | 51.61 | 20.96 | 38.84 |
| VIT16 | LIN. P. | 67.17 | 57.01 | 35.43 | 60.96 | 35.41 | 47.20 |
| | LIN. P. W/ PCA AND FASTICA | 68.12 | 58.45 | 38.41 | 63.89 | 37.17 | 49.48 |
| | LIN. P. W/ FASTICA | 67.96 | 58.38 | 38.75 | 65.45 | 38.28 | 50.22 |

Table 4: Quantitative results for 8-shot transfer learning and domain generalization by different methods. Lin. P. (Linear Probe).

| ENCODERS | METHODS | SOURCE IMAGENET | V2 | SKETCH | R | A | AVG. |
|---|---|---|---|---|---|---|---|
| RN50 | LIN. P. | 49.33 | 40.83 | 15.06 | 31.23 | 10.99 | 24.53 |
| | LIN. P. W/ FASTICA | 51.99 | 43.58 | 15.47 | 34.35 | 12.85 | 26.56 |
| | LIN. P. W/ PCA AND FASTICA | 51.42 | 42.93 | 17.28 | 35.53 | 12.33 | 27.02 |
| RN101 | LIN. P. | 55.41 | 46.04 | 23.38 | 43.26 | 16.88 | 32.39 |
| | LIN. P. W/ FASTICA | 56.59 | 47.47 | 22.09 | 44.59 | 18.39 | 33.14 |
| | LIN. P. W/ PCA AND FASTICA | 55.84 | 46.59 | 23.68 | 44.94 | 18.25 | 33.37 |
| VIT32 | LIN. P. | 55.17 | 46.11 | 25.53 | 45.32 | 18.35 | 33.83 |
| | LIN. P. W/ FASTICA | 56.90 | 47.96 | 27.62 | 49.13 | 20.31 | 36.26 |
| | LIN. P. W/ PCA AND FASTICA | 55.83 | 46.55 | 26.54 | 46.77 | 18.80 | 34.67 |
| VIT16 | LIN. P. | 61.82 | 52.34 | 32.26 | 55.93 | 32.63 | 43.29 |
| | LIN. P. W/ FASTICA | 63.55 | 54.81 | 34.21 | 61.54 | 38.21 | 47.29 |
| | LIN. P. W/PCA AND FASTICA | 63.47 | 54.32 | 35.83 | 61.88 | 37.35 | 47.36 |

Table 5: Quantitative results for 4-shot transfer learning and domain generalization by different methods. Lin. P. (Linear Probe).

| ENCODERS | METHODS | SOURCE IMAGENET | V2 | SKETCH | R | A | AVG. |
|---|---|---|---|---|---|---|---|
| RN50 | LIN. P. | 41.34 | 33.67 | 11.55 | 26.27 | 9.67 | 20.29 |
| | LIN. P. W/ FASTICA | 44.10 | 36.07 | 12.75 | 30.15 | 11.64 | 22.65 |
| | LIN. P. W/ PCA AND FASTICA | 42.86 | 35.38 | 12.29 | 28.81 | 9.79 | 21.57 |
| RN101 | LIN. P. | 48.23 | 39.53 | 18.80 | 38.10 | 14.32 | 27.69 |
| | LIN. P. W/ FASTICA | 49.43 | 41.02 | 17.49 | 39.33 | 15.25 | 28.27 |
| | LIN. P. W/ PCA AND FASTICA | 49.01 | 40.25 | 19.26 | 39.71 | 14.75 | 28.49 |
| VIT32 | LIN. P. | 47.82 | 39.53 | 21.51 | 40.94 | 15.99 | 29.49 |
| | LIN. P. W/ FASTICA | 49.43 | 40.66 | 22.66 | 41.78 | 16.41 | 30.38 |
| | LIN. P. W/ PCA AND FASTICA | 49.48 | 41.09 | 23.72 | 43.48 | 16.77 | 31.27 |
| VIT16 | LIN. P. | 54.30 | 46.06 | 27.58 | 50.76 | 29.24 | 38.41 |
| | LIN. P. W/ FASTICA | 56.65 | 48.18 | 28.27 | 55.50 | 33.39 | 41.33 |
| | LIN. P. W/ PCA AND FASTICA | 56.16 | 47.46 | 30.21 | 55.49 | 31.71 | 41.22 |

Table 6: Quantitative results for 1-shot transfer learning and domain generalization by different methods. Lin. P. (Linear Probe).

| | | SOURCE | TARGET (IMAGENET-) | | | | |
|---|---|---|---|---|---|---|---|
| ENCODERS | METHODS | IMAGENET | V2 | SKETCH | R | A | AVG. |
| RN50 | LIN. P. | 21.74 | 18.24 | 5.68 | 15.41 | 6.55 | 11.47 |
| | LIN. P. W/ FASTICA | 23.22 | 19.68 | 6.37 | 13.84 | 7.21 | 11.77 |
| | LIN. P. W/ FASTICA | 24.06 | 20.26 | 6.85 | 17.54 | 8.05 | 13.18 |
| RN101 | LIN. P. | 26.05 | 21.48 | 9.90 | 23.85 | 10.17 | 16.35 |
| | LIN. P. W/ FASTICA | 27.50 | 23.33 | 8.35 | 17.87 | 10.71 | 15.07 |
| | LIN. P. W/ PCA AND FASTICA | 28.50 | 24.17 | 11.63 | 26.38 | 12.28 | 18.62 |
| VIT32 | LIN. P. | 26.99 | 22.99 | 11.93 | 25.25 | 11.56 | 17.93 |
| | LIN. P. W/ FASTICA | 29.21 | 24.80 | 9.97 | 21.23 | 12.23 | 17.06 |
| | LIN. P. W/ PCA AND FASTICA | 29.05 | 24.45 | 12.39 | 27.61 | 12.56 | 19.25 |
| VIT16 | LIN. P. | 32.42 | 27.64 | 16.34 | 34.28 | 21.84 | 25.02 |
| | LIN. P. W/ FASTICA | 34.35 | 29.31 | 13.91 | 28.61 | 23.24 | 23.77 |
| | LIN. P. W/ PCA AND FASTICA | 35.20 | 30.26 | 19.17 | 38.87 | 26.41 | 28.68 |

## A.8 MORE RESULTS ON FEW-SHOT LEARNING TASK

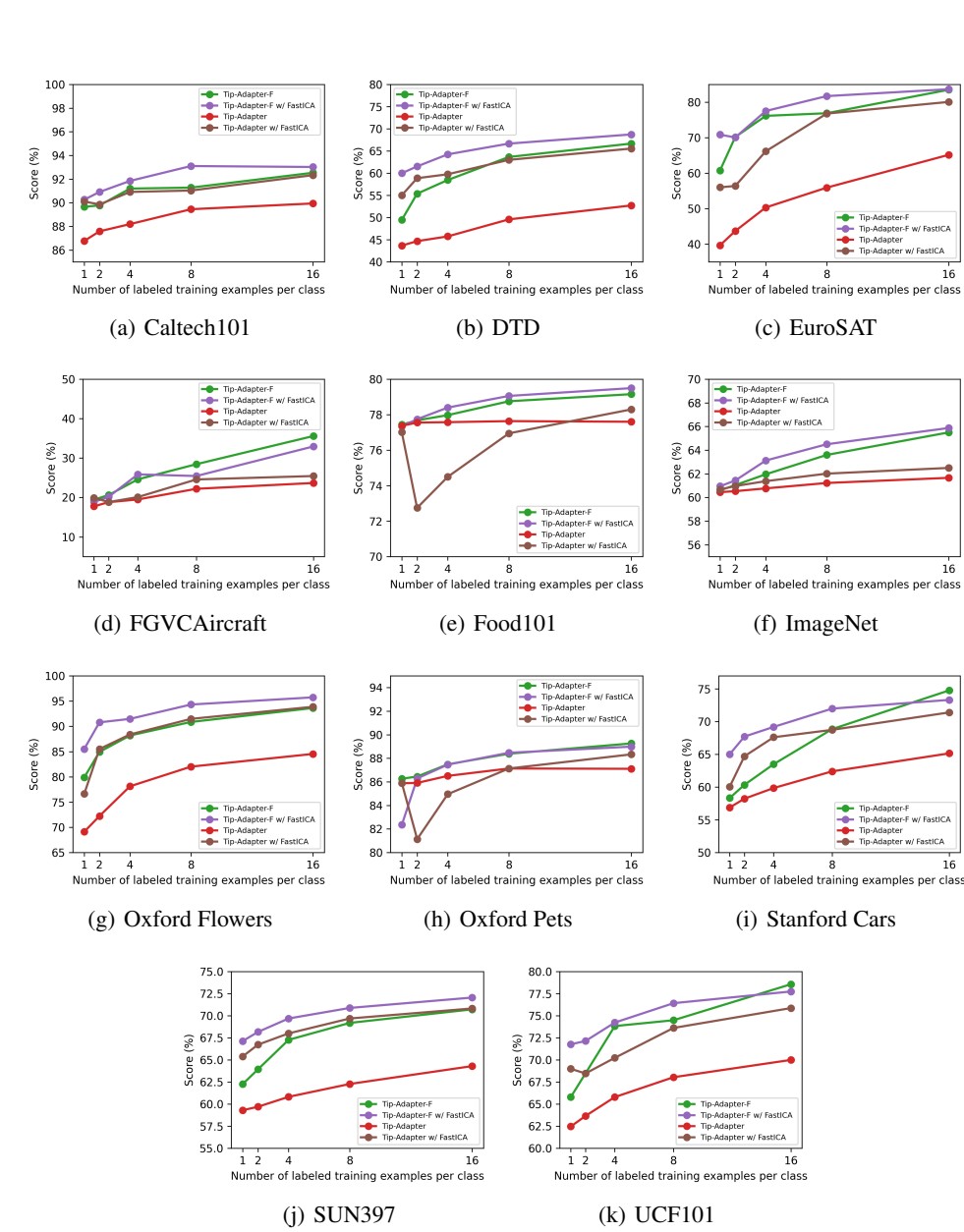

Figure 8: More results on few-shot learning task: A comparison of top-1 accuracy (%) achieved by various few-shot CLIP adaptation methods across 11 datasets. The x-axis indicates the number of training examples per class.The incorporation of FastICA notably enhances the performance of the original methods, Tip-Adapter and Tip-Adapter-F, proposed by Zhang et al. (2022a).

| |
|---|
| ReLU(BN(ConvTranspose2d(512, 512, kernelsize=1, stride=1, padding=0))) |
| ReLU(BN(ConvTranspose2d(512, 64, kernelsize=4, stride=1, padding=0))) |
| ReLU(BN(ConvTranspose2d(64, 64, kernelsize=4, stride=1, padding=0))) |
| ReLU(BN(ConvTranspose2d(64, 32, kernelsize=4, stride=1, padding=0))) |
| ReLU(BN(ConvTranspose2d(32, 32, kernelsize=4, stride=1, padding=0))) |
| ConvTranspose2d(32, 3, kernelsize=4, stride=2, padding=1) |

Table 7: Decoder for the image data.

## A.9 IMPLEMENTATION DETAILS

We perform all experiments using the GPU RTX 4090, equipped with 32 GB of memory.

**Synthetic Data**   We consider latent coupled variables $\mathbf{z}_x$ and $\mathbf{z}_t$, each with a dimensionality of 10. Additionally, we have modality-specific latent variables $\mathbf{m}_x$ and $\mathbf{m}_t$, both set to a dimension of 5. The process begins with sampling from the marginal distribution $p(\mathbf{z}_x)$, and the samples of modality-specific latent variables $\mathbf{m}_x$ and $\mathbf{m}_t$ are obtained by sampling from Gaussian distributions with zero mean and one variance. We then create real pairs by sampling from the conditional distribution $p(\mathbf{z}_t|\mathbf{z}_x)$. The observational data $\mathbf{x}$ and $\mathbf{t}$ are generated using two different Multi-Layer Perceptrons (MLPs). Specifically, we utilize MLPs comprising three hidden layers with leaky ReLU units and random weights. To ensure the invertibility of the MLP g, we carefully control the condition number of the weight matrices. For our feature encoders concerning both $\mathbf{z}_t$ and $\mathbf{z}_x$, we adopt an MLP architecture with leaky ReLU units.

**Disentangled Representation Learning on CelebA**   To obtain disentangled representations for the CelebA dataset, we initially employ the FastICA implementation available in the scikit-learn software on the features extracted from the pretrained ViT-B/32 encoder. Subsequently, we train the decoder, as outlined in Table 7, utilizing Mean Squared Error (MSE) loss.

**Experiments of Linear Probe**   In our experiments with ImageNet-Type data, we utilized the PCA and FastICA implementations provided by scikit-learn. For our proposed method, which combines PCA and ICA, we configured the number of components to 500 for PCA, and for FastICA, we set it to 160 for 1, 2, and 4-shot learning scenarios, and 200 for 8 and 16-shot learning scenarios. When employing ICA alone, we chose to use 300 components. For the proposed method with ICA only, we set number of components to 300. Following the setting of linear probe in CLIP, we train a logistic regression classifier using scikit-learn's L-BFGS implementation, with maximum 1,000 iterations. We determine the L2 regularization strength using a hyperparameter sweep on the validation sets over the range between $10^{-6}$ and $10^{6}$, with 96 logarithmically spaced steps. To save compute required for the sweeps, we perform a parametric binary search and iteratively halves the interval around the peak until it reaches a resolution of 8 steps per decade. The hyperparameter sweeps are performed on a validation split of each dataset.

**FastICA as a plug-and-play Tool.**   We incorporate FastICA in the framework proposed in Zhang et al. (2022a) to enhance its ability for few shot learning. The framework consists of two primary modules: one keeps the zero-shot capabilities of pre-trained CLIP, ensuring effective utilization of prior knowledge, while the other, the cache module, constitutes the central contribution of the work. The cache module endeavors to transfer knowledge from labeled training samples. Given the above, we integrate FastICA into the cache module, preserving the invaluable prior knowledge derived from the zero-shot abilities of pre-trained CLIP. For parameter settings in FastICA, we opted for 100 components for the majority of datasets. Specifically, we assigned 350 components for the ImageNet dataset, 300 components for the OxfordPets dataset, and 50 components for the EuroSAT dataset. A learning rate of 0.1 was employed for implementation. For the remaining parameter settings, we adhered to the specifications outlined by Zhang et al. (2022a).

