# OpenReview forum: "Beyond DAGs: A Latent Partial Causal Model for Multimodal Learning"
_ICLR.cc/2025/Conference — Submitted to ICLR 2025_

### Official Review · Reviewer_PGzN · 2024-10-30

**Soundness:** 3
**Presentation:** 2
**Contribution:** 2
**Rating:** 5
**Confidence:** 2

**Summary:**

Post rebuttal: I appreciate the author's rebuttal and discussion, which resolve some of my main concerns. I am raising score to 5 but lower my confidence score, because (1) I still think the paper is written in a very confusing way, which is very easy to distract the reader to access the main idea; (2) the theoretical results depend on the b of exact modality matching, which is impossible in practice. Therefore, I am not sure how significant the results are.

==================

This paper questions the DAG structure in multi-modal contrastive learning in real world problems, and propose a latent partial causal model for multimodal data. The partial causal structure can be more representative for latent representation relation modeling. The paper derives several theoretical results to help understanding the multi-modal contrastive loss, and validate the disentaglement capabilities of pretrained CLIP in learning disentangled representations.

**Strengths:**

1. The idea of latent partial causal modeling in multi-modal contrastive learning is interesting.
2. Some theoretical results on the multi-modal contrastive loss can help one better understand the underlying mechanism of multi-modal contrastive learning.

**Weaknesses:**

While the problem studied in the paper is interesting, I find the paper very hard to follow.

In my understanding, the paper proposes the latent partial causal structure in multi-modal contrastive learning. I thought there must be text to explain why this architecture is good, how to design algorithm algorithm to optimize the model and why properties this architecture could have. Unfortunately, I cannot catch what the paper is presenting. More specifically, Section 3.1 talks about the latent partial causal structure, then Section 3.2 directly analyze the standard multi-modal contrastive learning objective used in for example CLIP. But how is the objective related to your model? I believe the latent partial causal model should have a different objective because it has extra latent variables. Also, in Section 4, Theorem 4.1, I believe h is the composition of f_x and g_x, but why it only takes z_x (or z_t) as input? Where is m_x? And I did not follow the insights explained for Theorem 4.1. My concerns apply to Section 4.2.

Finally, I am not sure what the experiments try to tell us. There is not enough information to explain the results. For example, Table 1 tries to evaluate the identificabiulity of the model, but how? and how do you calculate the recovered \hat{z}_x? For the rest of the experiments, I think they are quite toy experiments, and I am not convinced about the advantage of the proposed method from the reported results.

**Questions:**

See the weakness part.

---

> ### Author Response · Authors · 2024-11-21
> **Response**
>
> Thank you for your thoughtful and detailed feedback. We greatly appreciate your recognition of the potential impact of our approach, as well as your constructive suggestions for improving our work.
>
> ____
>
> **Q1:** I thought there must be text to explain why this architecture is good, how to design algorithm to optimize ... But how is the objective in Section 3.2  related to your model in Section 3.1?...
>
> **R1:** As mentioned in the introduction, assuming a DAG (Directed Acyclic Graph) structure can be a strong constraint compared to non-DAG models, as it requires an assumption about the causal direction. Additionally, identifying latent DAGs often necessitates various assumptions. In light of this, we pose the question: is a non-DAG assumption sufficient for certain applications? To address this, we propose a latent partial causal model depicted in Figure 1, which has the flexibility to encompass multiple possible DAG structures, as illustrated in Figure 2. Using this partial causal generative model, we examine a multimodal contrastive learning framework (inference model) to infer latent variables, given its success in various applications, such as CLIP. Alternative inference models, such as those maximizing likelihood, also exist. By applying certain assumptions to the proposed latent partial causal models, we prove that multimodal contrastive learning can identify latent coupled variables (e.g., relation between multimodal contrastive objective and the proposed model), as demonstrated in Theorem 4.1 and Corollary 4.2 on a hypersphere, and Theorem 4.3 and Corollary 4.4 on convex bodies. These theoretical results indicate that multimodal contrastive learning identifies latent coupled variables, and, if we assume these latent coupled variables are independent, multimodal contrastive learning can further identify them component-wise. Combined with linear ICA, this corresponds to achieving disentanglement.
>
> **Q2:** Also, in Section 4, Theorem 4.1, I believe h is the composition of f_x and g_x, but why it only takes z_x (or z_t) as input? Where is m_x? And I did not follow the insights explained for Theorem 4.1. My concerns apply to Section 4.2.
>
> **R2:** As we analysed in Theorem 4.1 and Theorem 4.3,the optimal normalized multimodal contrastive loss defined by Eq. (2) will effectively remove the influence of $\mathbf{m}_x$ in $\mathbf{h}$. Intuitively, for paired images and text, the contrastive loss functions to minimize the distance between their respective representations. In Figure 1, we can see that $\mathbf{m}_t$ and $\mathbf{m}_x$ are independent variables. If these components are not accounted for in the representations, the distance between the paired image and text representations will be influenced by $\mathbf{m}_t$ and $\mathbf{m}_x$. Since $\mathbf{m}_t$ and $\mathbf{m}_x$ are independent and can vary randomly, this introduces variability that makes it impossible to control the distance between the paired representations. This variability contradicts the goal of contrastive loss, which is to minimize the distance between matched image and text representations effectively. For more rigorous derivation, please refer to Eqs. 18 and 19.
>
> **Q3:** Table 1 tries to evaluate the identificabiulity of the model, but how?
>
> **R3:** Table 1 results is to verify our identifbaility analysis, e.g., on hyperspehre, latent coupled variables $\mathbf{z}_x$ are identified up to linear transformation, e.g., Theorem 4.1 and Corollary 4.2, on convex, latent coupled variables $\mathbf{z}_x$ are identified up to permutation , e.g., Theorem 4.3 and Corollary 4.4. Therefore, on hypersphere, we use R2, which quantifies the degree of any linear correlation between the estimated $\mathbf{\hat z}_x$ (representations obtained by multimodal contrastive loss) and the true $\mathbf{z}_x$, for eluviation. On the convex bodies, we use the MCC, a standard method for evaluating the recovery of independent components in nonlinear ICA, to eluviate that latent coupled variables $\mathbf{z}_x$ are identified up to permutation.
>
> **Q4:** I think they are quite toy experiments, and I am not convinced about the advantage of the proposed method from the reported results
>
> **R4** This work does not propose a new method; instead, it provides an identifiability analysis for multimodal contrastive learning. As Theorem 4.1 and Corollary 4.2 demonstrate, multimodal contrastive learning has the ability to identify latent coupled variables up to a linear transformation. Thus, using linear ICA allows for component-wise recovery of latent coupled variables, which shows the potential for disentanglement in theory. To align with these theoretical results, we use pre-trained CLIP, followed by linear ICA, to verify our theoretical claims in epxeirments. In summary, we are not introducing a new method or verifying the advantages of the new method; rather, all our experiments are designed to align with and validate our theoretical claims.

---

> > ### Author Response · Authors · 2024-11-21
> >
> > ----
> >
> > Dear reviewer PGzN,
> >
> > Thank you sincerely for investing your time and consideration into this evaluation process. Our aim in providing this clarification is to address your concerns, and we hope that these responses will help you gain a deeper understanding of our work. Your thoughtful re-evaluation holds immense significance for us, and we sincerely hope that our clarification proves instrumental in facilitating a comprehensive and constructive reassessment. If there is anything that remains unclear or continues to be a concern, please let us know, and we will be happy to provide further responses.

---

> > > ### Comment · Reviewer_PGzN · 2024-11-26
> > >
> > > Thanks for the rebuttal. However, the explanation still does not resolve my questions:
> > >
> > > 1. What is the objective function of the proposed latent partial causal model in Figure 1? How is it related to the standard multi-modal contrastive loss?
> > > 2. The h function in eq.6, I believe it is the composition of f_x and g_x,, but why does it only have z_x/z_t as input? I believe the input should also include m_x/m_t., as shown in eq.18 and eq.19.

---

> ### Author Response · Authors · 2024-11-26
>
> Dear Reviewer PGzN,
>
> Thank you for your reply and the opportunity for further discussion.
>
> ---
>
> Before responding, we would like to provide some background to help the reviewer better understand our work.
>
> One of our main contributions is explaining the success of multimodal contrastive learning. To achieve this, we propose investigating multimodal contrastive representation learning from a latent causal perspective. This approach allows us to examine high-level latent causal factors shared across modalities.
>
> To this end, we first introduce a latent causal model assumption for the generative process of multimodal datasets. This provides a framework to interpret how such multimodal data are generated. Specifically, we propose our latent partial causal model, illustrated in Figure 1. Using this model, we analyze how the multimodal contrastive loss can be connected to the underlying latent causal model depicted in Figure 1.
>
> Motivated by the insights in Section 3.2, we hypothesize that the representations learned via multimodal contrastive loss may identify latent coupled variables $z_x$ and $z_t$ within the proposed latent partial causal model. For instance, the learned representation $\hat z_x$ by multimodal contrastive loss could belong to the same equivalence class as the true latent variable $z_x$.
>
> To rigorously establish this result, we introduce additional assumptions about the proposed latent partial causal model. For example, these include conditions such as Eq. (4) for hyperspheres or Eq. (7) for convex bodies. Under these assumptions, we prove that the features learned by the multimodal contrastive loss are indeed related to the true latent variables z_x and z_t. This is formalized in Theorem 4.1 and Corollary 4.2 for hyperspheres,  which establish that $\hat z_x$= $A z_x + c$ , where A is a linear matrix, c is a constant. Similarly, Theorem 4.3 and Corollary 4.4 extend this result to convex bodies, showing that $\hat z_x$= $P z_x + c$, where P is a permutation matrix with scaling.
>
> These results demonstrate that the features learned by multimodal contrastive loss is related with essential, generalizable patterns in the data, such as high-level latent coupled variables (e.g., $z_x$). This relationship provides theoretical support for the success of multimodal contrastive learning.
>
> Furthermore, since we assume that $z_x$ has independent components, our identifiability analysis on hyperspheres (e.g., Theorem 4.1 and Corollary 4.2) demonstrates that for the learned representation $\hat z_x$, we have $\hat z_x = A z_x + c$. This result enables the application of linear ICA on $\hat z_x$ to recover the independent components of $z_x$ , coresponding to the potential for disentanglement.
>
> We hope the above background helps the reviewer gain a deeper understanding of our work.
>
>
>
> --------
>
> For your concerns:
>
> 1. As mentioned in Introduction (e.g., lines 061–062), 'Given the proposed.., we analyze it within the widely used multimodal contrastive learning paradigm (e.g., inference).' Specifically, we analyse the proposed model using the multimodal contrastive loss defined in Eq. (1), which belongs to a broader class of multimodal contrastive losses, including multimodal contrastive loss on hypersphere.
>
> 2. Indeed, h_x is a composition of f_x and g_x. Without any further analysis, it would naturally include both z_x and m_x. However, by considering the optimal solution of the multimodal contrastive loss and the independence properties in the proposed latent partial causal model, e.g., m_x is independent z_x, it can be shown that h_x do not depend on m_x.  As we metioned in **R2**, for a more rigorous derivation, please refer to Eqs. 18 and 19, which demonstrate that the partial derivative of h_x with respect to m_x is equal to 0, implying that h_x does not depend on m_x. (similarly for h_t). Additionally, we provide an intuitive explanation in **R2**.

---

> > ### Author Response · Authors · 2024-11-29
> > **Follow-Up on Discussion Phase**
> >
> > Dear Reviewer PGzN,
> >
> > As the discussion phase is nearing its conclusion, we kindly ask if your concerns have been adequately addressed. We understand you may have a busy schedule, but if you have any follow-up questions or remaining concerns, please let us know.
> >
> > If our rebuttal has resolved your concerns, we would greatly appreciate it if you could update your score and share your feedback.
> >
> > Thank you for your time and effort!
> >
> > Best,
> >
> > Authors.

---

> > > ### Comment · Reviewer_PGzN · 2024-12-02
> > >
> > > Thanks for the clarification, which resolves some of my concerns, especially for my first question. I would highly encourage the authors to make this explanation clear in the revision.
> > >
> > > However, for my second question, after checking the proof for Lemma 5.1, I am afraid I cannot agree the proof. Specifically,  the first sentence of the proof:
> > >
> > > "for pair (mx, zx), (mt, zt), by differentiating it with respect to mx, we have eq.18, due to the independence between mx and (mt, zt). "
> > >
> > > I am afraid I don't agree with this step, because hx(mx, zx) = ht(mt, zt) does not necessarily imply \nabla_mx hx(mx, zx) = \nabla_mx ht(mx, zx). Thus, I highly doubt the theoretical results in this paper.

---

> > > > ### Author Response · Authors · 2024-12-02
> > > >
> > > > Thank you for your additional concerns.
> > > >
> > > > Please note that **$h_x(m_x, z_x) = h_t(m_t, z_t)$ almost surely for every paired $((m_x,z_x),(m_t, z_t))$**. This follows from the continuity of $h_x(m_x, z_x)$ and  $h_t(m_t, z_t)$ which ensures that the equality holds in a small neighborhood around $m_x$ . Consequently, the gradients on both sides must also match.
> > > >
> > > > Similar results, where $m_t$ and $m_x$ can be removed from $h$ have already been proven. Please refer to Daunhawer, Imant, et al., Identifiability Results for Multimodal Contrastive Learning, arXiv preprint arXiv:2303.09166 (2023)."

---

> > > > > ### Author Response · Authors · 2024-12-02
> > > > > **Could you kindly verify if the provided clarification addresses your concerns?**
> > > > >
> > > > > Dear Reviewer PGzN,
> > > > >
> > > > > Could you kindly verify if the clarifications we provided adequately address your concerns? Your feedback is invaluable to us, and we want to ensure we’ve addressed your comments appropriately before the discussion phase concludes.
> > > > >
> > > > > Thank you for your time and consideration. We truly appreciate your insights and guidance.
> > > > >
> > > > > Best regards,
> > > > >
> > > > > Authors

---

> > > > > ### Comment · Reviewer_PGzN · 2024-12-02
> > > > >
> > > > > Can you point me to where the reference paper did similar thing as what you did?

---

> ### Author Response · Authors · 2024-12-03
>
> Thank you for your further concern.
>
> Before responding, we re-reviewed our process to better understand your concerns. During this review, we noted your previous comment: "because because hx(mx, zx) = ht(mt, zt) does not necessarily imply \nabla_mx hx(mx, zx) = \nabla_mx ht(mx, zx)". We are unsure whether this is a typo. If "\nabla_mx ht(mx, zx)" is not a typo, please note that ht corresponds to the encoder for text data, and thus the effective input should be (mt, zt), i.e., "\nabla_mx ht(mx, zx)"should be "\nabla_mx ht(mt, zt) ".  If it is a typo, please refer to the below.
>
> -------
>
>
> Consider the paired data $((m_x, z_x), (m_t, z_t))$, the complete expression of $h_x(m_x, z_x)=h_t(m_t, z_t)$ should be $h_x((m_x, z_x), (m_t, z_t))=h_t((m_x, z_x), (m_t, z_t))$. This is because during the training process in multimodal contrastive learning, two different encoders are applied to two distinct modalities. As a result, in $h_x((m_x, z_x), (m_t, z_t))$, $(m_t, z_t)$ term is masked. Similarly for $h_t((m_x, z_x), (m_t, z_t))$. We now proceed to prove that their derivatives are equal everywhere.
>
>
> The derivative of  $ \frac {\partial {h} _ x( {m} _ x,{z} _ x, m_t, z_t )} {\partial {m}_x}$ is defined as:
>
>  $ \frac {\partial {h} _ x({m} _ x,{z} _ x, m_t, z_t )} {\partial {m}_x} = \lim _ {e \rightarrow 0}    \frac {{h} _ x({m} _ x + e,{z} _ x, m_t, z_t)- {h} _ x({m} _ x,{z} _ x, m_t, z_t) } {e}  $ .
>
> Similarly, the derivative of  $ \frac {\partial {h} _ t( {m} _ x,{z} _ x, {m} _ t,{z} _ t)} {\partial {m}_x}$ is defined as:
>
> $ \frac {\partial {h} _ t({m} _ x,{z} _ x, {m} _ t,{z} _ t)} {\partial {m}_x} = \lim _ {e \rightarrow 0}    \frac {{h} _ t( {m} _ x + e,{z} _ x, {m} _ t,{z} _ t)- {h} _ t({m} _ x,{z} _ x, {m} _ t,{z} _ t) } {e}  $ .
>
> Since $h_x(m_x, z_x, m_t, z_t)=h_t(m_x, z_x, m_t, z_t)$,  and considering that we define paired data based on z as $p(z_t|z_x)$ in Eq. (4) (or (7)), the pairing $ ({m} _ x + e,{z} _ x, m_t, z_t)$ holds despite the disturbance
> e on m_x. Consequently, ${h} _ x({m} _ x + e,{z} _ x, m_t, z_t) = {h} _ t({m} _ x + e,{z} _ x, {m} _ t,{z} _ t)$.
>
> Given the above,
> $\frac {\partial {h} _ x({m} _ x,{z} _ x, m_t, z_t )} {\partial {m}_x} = \lim _ {e \rightarrow 0}    \frac {{h} _ x({m} _ x + e,{z} _ x, m_t, z_t)- {h} _ x({m} _ x,{z} _ x, m_t, z_t) } {e} = \lim _ {e \rightarrow 0}    \frac {{h} _ t({m} _ x + e,{z} _ x, m_t, z_t)- {h} _ t({m} _ x,{z} _ x, m_t, z_t) } {e} =    \frac {\partial {h} _ t({m} _ t,{z} _ t, {m} _ x,{z} _ x )} {\partial {m}_x}$ (Here $m _ x \perp m _ t, z _ t$, so that $\frac {\partial {h} _ t({m} _ t,{z} _ t, {m} _ x,{z} _ x )} {\partial {m}_x}=\frac {\partial {h} _ t({m} _ t,{z} _ t )} {\partial {m}_x}=0$). Therefore, the partial derivatives are equal.
>
> For the reference, please see Eq. (16), which indicates that if the function h, analogous to h in our work, depends on $m_1$ or $m_2$ (corresponding to our m_x and m_t), then $|h_1-h_2|>0$, h_1 and h_2 are analogous to h_x and h_t in our work. In other words, h cannot depend on modality-specific information such as $m_1$ or $m_2$, to satisfy $h_1= h_2$.

---

> > ### Comment · Reviewer_PGzN · 2024-12-03
> >
> > Thanks for the explanation, which seems to resolve my concern on this issue.  I raise my score to 5 but lower my confidence score, because (1) I still think the paper is written in a very confusing way, which is very easy to distract the reader to access the main idea; (2) the theoretical results depend on the b of exact modality matching, which is impossible in practice. Therefore, I am not sure how significant the results are.

---

> ### Author Response · Authors · 2024-12-03
>
> Thank you sincerely for dedicating your time to reviewing our work and, most importantly, for meticulously checking our proof step by step.
>
> We would like to highlight two key contributions of our work (More detailed contributions are summarized in the Introduction section): 1) Unlike traditional approaches relying on DAGs, we demonstrate that latent causal models with non-DAG assumptions among latent causal variables are sufficient for certain applications, such as explaining the success of multimodal contrastive learning. 2) To the best of our knowledge, this is the first work to provide theoretical guarantees for the potential of disentanglement in models trained via multimodal contrastive learning, thereby pushing the boundaries of pre-trained models like CLIP.
>
> The condition of exact modality matching in our theoretical results is a commonly used assumption in previous works analyzing multimodal data, e.g., [1, 2]. Relaxing this condition represents a non-trivial and compelling direction for future research.
>
> [1] Daunhawer, Imant, et al. "Identifiability results for multimodal contrastive learning." ICLR (2023).
>
> [2] Yao, Dingling, et al. "Multi-view causal representation learning with partial observability." ICLR (2004).

---

### Official Review · Reviewer_9Eyo · 2024-11-02

**Soundness:** 3
**Presentation:** 3
**Contribution:** 2
**Rating:** 6
**Confidence:** 3

**Summary:**

The paper extends the identifiability results from previous work [1] to the case of multimodal contrastive learning. To validate their theoretical results, the authors show that shared information between modalities can be recovered from the representations obtained with a multimodal contrastive learning objective.  In addition, they demonstrate that their results are satisfied empirically even when certain assumptions are violated.  Finally, from their theoretical insights, the authors postulate that CLIP, which is trained with a multimodal contrastive learning objective, produces representations that can be well-disentangled in separate factors. This last claim is validated by applying ICA on top of the CLIP embeddings and showing positive results for few-shot classification and domain generalization.

[1] Zimmermann et al. Contrastive Learning Inverts the Data Generating Process, ICML, 2021.

**Strengths:**

- To my understanding, the paper presents a solid theoretical analysis for identifiability in multimodal contrastive learning.
- Empirical results validate the authors' theoretical insights
- The insight that ICA can improve the performance for robust few-shot learning with CLIP representations is useful and interesting

**Weaknesses:**

- The paper is strongly based on the results presented in [1] and provides similar insights as previous works [2,3], which limits the significance and novelty of the presented results.
- While "going beyond DAGs" is central in the title, to me it does not seem like the paper focuses on possible advantages of "non-DAG assumptions". To my understanding, the paper extends the identifiability results for multi-view contrastive learning from [1] to the multimodal contrastive learning case. Could the authors better clarify the centrality of the non-DAG assumption and why it is necessary for the insights in this work?
-  The authors state "Informed by our identifiability results and the empirical evidence presented earlier, we have grounds to claim that the pre-trained CLIP model possesses disentanglement ability." I do not immediately understand what is the rationale behind this claim. Given the presented insights, one can claim that shared information can be disentangled from modality-specific information in the latent space, but why should individual factors (i.e. individual attributes in celebA) be disentangled?



[1] Zimmermann et al. Contrastive Learning Inverts the Data Generating Process, ICML, 2021.
[2] Daunhawer et al. Identifiability results for multimodal contrastive learning. arXiv preprint arXiv:2303.09166, 2023.
[3] Yao et al. Multi-view causal representation learning with partial observability. arXiv preprint arXiv:2311.04056, 2023.

**Questions:**

See Weaknesses

---

> ### Author Response · Authors · 2024-11-21
> **Response**
>
> Thank you for your thoughtful and detailed feedback. We appreciate your recognition of the potential impact of our approach and your constructive suggestions for improving our work.
>
> ------
>
> **Q1:** The paper is strongly based on the results presented in [1] and provides similar insights as previous works [2,3],
>
> **R1:**  One of our primary motivations is to explore whether partial causal models are sufficient for specific applications. To address this, we introduce latent partial causal models and use multimodal contrastive learning as an illustrative example. We demonstrate that: (1) partial causal models can theoretically support the success of multimodal contrastive learning by enabling the identification of latent coupled variables, and (2) these models can also facilitate the disentanglement potential of multimodal contrastive learning. Our findings suggest that partial causal models may be adequate for certain applications, offering a novel insight: full DAG assumptions in latent causal analysis might be unnecessary for some practical cases. This avoids the restrictive and often impractical assumptions associated with full DAGs, an aspect previously overlooked. Additionally, while some results extend single-modal analysis from [3], our contributions include: (1) the development of Theorems 4.1 and 4.2, bridging the theoretical gap between single-modal and multimodal analysis, showing the applicability of certain results from [3] in a multimodal context, as elaborated in subsequent paragraphs, and (2) highlighting the underexplored disentanglement potential of pre-trained multimodal models like CLIP. This potential, achievable through methods as straightforward as linear ICA, warrants consideration for future applications
>
> **Q2:** Could the authors better clarify the centrality of the non-DAG assumption and why it is necessary for the insights in this work?
>
> **R2:** The main motivation of this work is that, in general, causal analysis requires satisfying DAG (Directed Acyclic Graph) assumptions. For example, when modeling multimodal data, there are three possible DAG structures, as shown in Figure 2. Each DAG requires specific assumptions, such as defining the causal relationship between zx and zt. Additionally, identifying these latent causal variables and the DAG structure typically demands further assumptions, such as specific parametric constraints on the variables or limitations on the function class describing the causal relationships. However, verifying these assumptions (including graph structure and parametric or function class constraints) is often difficult in real-world applications.
>
> Motivated by this challenge, a natural question arises: are DAG constraints always necessary? Could non-DAG causal models be sufficient for certain applications? To address this, we propose the latent partial causal model shown in Figure 1, in which a undirected edge between $\mathbf{z}_x$ and $\mathbf{z}_t$ is provided so that we avoid further assumptions on the causal direction between $\mathbf{z}_x$ and $\mathbf{z}_t$. By combining this partial model with multimodal contrastive learning as an example, we demonstrate that multimodal contrastive learning can identify latent coupled variables in the proposed latent partial causal model, $\mathbf{z}_x$ and $\mathbf{z}_t$, supporting the effectiveness of multimodal contrastive learning and revealing its theoretical potential for disentanglement.
>
> These findings indicate that, for certain applications, non-DAG assumptions can be sufficient, removing the need for stringent DAG constraints. This insight is novel and significant, as it shows that the strict requirements of traditional DAG-based causal analysis may not always be necessary, thus broadening the scope of practical causal modeling.

---

> > ### Author Response · Authors · 2024-11-21
> >
> > **Q3** Informed by our identifiability results and the empirical evidence presented earlier, we have grounds to claim that the pre-trained CLIP model possesses disentanglement ability. I do not immediately understand what is the rationale behind this claim.
> >
> > **R3** Theorem 4.1 indicates that Eq. (5) represents a variant of the multimodal contrastive loss. As discussed in the paragraph following Theorem 4.1, minimizing Eq. (5) enables the identification of the latent variables $\mathbf{z}_x$ and $\mathbf{z}_t$ up to a linear transformation. This implies that the estimated features, e.g., embeddings, $\mathbf{\hat z}_x$ obtained from minimizing Eq. (5) are linearly related to the true $\mathbf{z}_x$, such that $\mathbf{\hat z}_x = \mathbf{A}\mathbf{z}_x+\mathbf{c}$. Consequently, linear ICA can be used on the features $\mathbf{\hat z}_x$ to simplify
> > $\mathbf{A}$ to a permutation matrix $\mathbf{P}$, resulting in new features $\mathbf{\hat z'}_x = \mathbf{P}\mathbf{z}_x$. This implies that as an equivalent loss as in Eq. (5), multimodal contrastive loss, can obtain features $\mathbf{\hat z}_x $ first, and then obtain independent latent components in $\mathbf{\hat z'}_x$
> > through linear ICA. This is also why we show individual factors. Therefore, models trained with multimodal contrastive loss, such as CLIP, inherently possess the potential for disentanglement. This result differs from previous works that may only disentangle $\mathbf{z}_x$ from $\mathbf{m}_x$, where $\mathbf{z}_x$ is only identified up to a nonlinear invertible mapping. In contrast, our identifiability results are more fine-grained, as they identify $\mathbf{z}_x$ up to component-wise permutation by using linear ICA.
> >
> >
> > Thank you for this question, which made us realize the need for a clearer explanation. We have updated the relevant sections in the new version, with the changes highlighted in blue, to further clarify this point.

---

> > > ### Author Response · Authors · 2024-12-02
> > > **Follow-Up on Addressing Your Concerns**
> > >
> > > Dear Reviewer 9Eyo,
> > >
> > > As the discussion phase deadline approaches, we kindly ask if you could let us know whether our response has addressed the concerns you raised. Your confirmation would mean a lot to us, as it will help ensure we’ve fully understood and addressed your valuable feedback.
> > >
> > > Thank you so much for your time and support throughout this process. We greatly appreciate your thoughtful input.
> > >
> > > Best regards,
> > >
> > > Authors

---

### Official Review · Reviewer_B7j4 · 2024-11-02

**Soundness:** 3
**Presentation:** 3
**Contribution:** 3
**Rating:** 6
**Confidence:** 3

**Summary:**

The paper introduces a latent partial causal model for multimodal data that captures the undirected relationships between the two latent coupled variables corresponding to each modality. Further, the author provides identifiability guarantees for the commonly used multimodal contrastive learning loss and demonstrates its potential for disentanglement with CLIP embedding on real-world images.

**Strengths:**

- The paper focuses on multimodal learning problems and provides a novel perspective on the latent partial causal model setting.
- The paper provides identifiability analysis for the provided causal model on both hypersphere and convex body, and demonstrates its disentanglement implication.
- Empirical results on a synthetic setting and CLIP embeddings support the theoretical results.

**Weaknesses:**

- Section 3.2 about prior matching and information preservation shares a lot in common with many concepts being explored in self-supervised learning at the high level, for example, the alignment-uniformity perspective in [1], and the mutual information perspective in [2] (given that mutual information can be decomposed into the difference between an entropy and a conditional entropy term, corresponding to matching and information preservation). The paper lacks discussions/comparisons with these perspectives.
- The condition in Theorem 4.1, $\mathbf{f}_x \circ \mathbf{g}_x=\mathbf{f}_t \circ \mathbf{g}_t$, looks strong. It could be hard to achieve in real-world data.
- Compared to unimodal representations from SSL methods and the analysis in [3], it is unclear what specific gain can be attained from the multimodal representations.

[1] Understanding Contrastive Representation Learning through Alignment and Uniformity on the Hypersphere.

[2] Representation Learning with Contrastive Predictive Coding.

[3] Contrastive Learning Inverts the Data Generating Process.

**Questions:**

- In Equation 5 of Theorem 4.1, is $z_t$ in $\mathbb{E}_{z_x\sim p(z_x)} [H(p(z_t|z_x),q_h(z_t|z_x))]$ the one corresponds to the current $z_x$, or it is from a random distribution $p(z_t)$?
- Can the authors give some intuition on how the modality-specific variables $m_x$ and $m_t$ are got rid of in Theorem 4.1 and how different  $m_x$ and $m_t$ would affect the results? Intuitively, when $m_x$ and $m_t$ dominate, it is harder to identify the latent partial causal variables.
- What is the reasoning behind the conclusion that Theorem 4.1 implies the potential for disentanglement of pretrained model embeddings?

---

> ### Author Response · Authors · 2024-11-21
> **Response**
>
> We thank Reviewer B7j4 for recognizing its identifiability analysis, disentanglement implication, a novel perspective on the latent partial causal model setting. We address the reviewer’s questions below.
>
> -------
>
> **Q1:** Section 3.2 about prior matching and information preservation shares a lot in common with many concepts being explored in  [1], and the mutual information perspective in [2]... The paper lacks discussions/comparisons with these perspectives.
>
> **R1:** Indeed, the alignment-uniformity perspective and mutual information (information preservation) have been discussed in prior work within the context of single-modal learning. However, these two aspects are typically investigated separately. In this work, we extend these perspectives to a multi-modal setting and, more importantly, combine them to explore their role in solving inverse problems. This combined way provides a new insight: multimodal contrastive learning has the potential to address inverse problems. Thank you for the suggestion. We have updated the paragraph on line 284 to include a clear comparison and emphasize our novel insight into the relationship between contrastive learning and solving inverse problems.
>
> **Q2:** The condition in Theorem 4.1, looks strong. It could be hard to achieve in real-world data.
>
> **R2:** We agree that this condition stems from theoretical analysis, which does not take optimization challenges into account. Achieving such conditions in real-world applications can be difficult, particularly due to issues like local optimization or limited real dataset. However, our primary theoretical focus is on identifiability results, where it is generally assumed that global optimization can be achieved. Further theoretical analysis related to optimization is indeed challenging and beyond the scope of this paper.
>
> **Q3:** Compared to unimodal representations from SSL methods and the analysis in [3], it is unclear what specific gain can be attained from the multimodal representations.
>
> **R3:** Nice question. One of our primary motivations is to explore whether partial causal models are sufficient for specific applications. To address this, we introduce latent partial causal models and use multimodal contrastive learning as an illustrative example. We demonstrate that: (1) partial causal models can theoretically support the success of multimodal contrastive learning by enabling the identification of latent coupled variables, and (2) these models can also facilitate the disentanglement potential of multimodal contrastive learning. Our findings suggest that partial causal models may be adequate for certain applications, offering a novel insight: full DAG assumptions in latent causal analysis might be unnecessary for some practical cases. This avoids the restrictive and often impractical assumptions associated with full DAGs, an aspect previously overlooked. Additionally, while some results extend single-modal analysis from [3], our contributions include: (1) the development of Theorems 4.1 and 4.3, bridging the theoretical gap between single-modal and multimodal analysis, showing the applicability of certain results from [3] in a multimodal context, as elaborated in subsequent paragraphs, and (2) highlighting the underexplored disentanglement potential of pre-trained multimodal models like CLIP. This potential, achievable through methods as straightforward as linear ICA, warrants consideration for future applications.
>
> **Q4** Can the authors give some intuition on how the modality-specific variables are got rid of in Theorem 4.1 .....
>
> **R4** This is also a good question. Let's consider the generative process illustrated in Figure 1. Intuitively, for paired images and text, the contrastive loss functions to minimize the distance between their respective representations. In Figure 1, we can see that $\mathbf{m}_t$ and $\mathbf{m}_x$ are independent variables. If these components are not accounted for in the representations, the distance between the paired image and text representations will be influenced by $\mathbf{m}_t$ and $\mathbf{m}_x$. Since $\mathbf{m}_t$ and $\mathbf{m}_x$ are independent and can vary randomly, this introduces variability that makes it impossible to control the distance between the paired representations. This variability contradicts the goal of contrastive loss, which is to minimize the distance between matched image and text representations effectively. For the case of where $\mathbf{m}_t$ and $\mathbf{m}_x$ dominate, we believe that the theoretical results should still hold. However, there may be significant challenges in optimization during implementation. For instance, the features learned by multimodal contrastive loss may incorporate both $\mathbf{m}_t$ and $\mathbf{m}_x$ information to some extent. The degree to which this information is included largely depends on the optimization process.

---

> > ### Author Response · Authors · 2024-11-21
> >
> > **Q5** What is the reasoning behind the conclusion that Theorem 4.1 implies the potential for disentanglement of pretrained model embeddings?
> >
> > **R5** Theorem 4.1 indicates that Eq. (5) represents a variant of the multimodal contrastive loss. As discussed in the paragraph following Theorem 4.1, minimizing Eq. (5) enables the identification of the latent variables $\mathbf{z}_x$ and $\mathbf{z}_t$ up to a linear transformation. This implies that the estimated features, e.g., embeddings, $\mathbf{\hat z}_x$ obtained from minimizing Eq. (5) are linearly related to the true $\mathbf{z}_x$, such that $\mathbf{\hat z}_x = \mathbf{A}\mathbf{z}_x+\mathbf{c}$. Consequently, linear ICA can be used on the features $\mathbf{\hat z}_x$ to simplify
> > $\mathbf{A}$ to a permutation matrix $\mathbf{P}$, resulting in new features $\mathbf{\hat z'}_x = \mathbf{P}\mathbf{z}_x$. This implies that as an equivalent loss as in Eq. (5), multimodal contrastive loss, can obtain features $\mathbf{\hat z}_x $ first, and then obtain independent latent components in $\mathbf{\hat z'}_x$
> > through linear ICA. Therefore, models trained with multimodal contrastive loss, such as CLIP, inherently possess the potential for disentanglement. We have updated the relevant paragraphs in the new version, with the changes highlighted in blue, to clarify this point further.
> >
> > **Q6** In Equation 5 of Theorem 4.1, is $z_t$...
> >
> > **R6**  $z_t$ is obtained by first sampling $z_x$, and then sampled from condition distribution $p(z_t|z_x)$ or $q_h(z_t|xz_x)$

---

> ### Comment · Reviewer_B7j4 · 2024-11-28
>
> Thank the authors for providing detailed clarifications, and I would like to maintain my positive score.

---

> > ### Author Response · Authors · 2024-11-28
> >
> > Dear Reviewer B7j4,
> >
> > Thank you sincerely for dedicating your time and for maintaining a positive opinion of our work. We deeply appreciate your thoughtful feedback and efforts in helping us improve this draft.
> >
> > Best,
> >
> > Authors

---

### Official Review · Reviewer_KBLq · 2024-11-03

**Soundness:** 2
**Presentation:** 3
**Contribution:** 2
**Rating:** 6
**Confidence:** 4

**Summary:**

The paper proposes a new model for understanding how different types of data (e.g., text and images) relate to each other, moving away from the usual directed acyclic graph (DAG) approach. Instead, it uses a structure with latent variables connected by undirected edges, which captures shared information between data types under different scenarios. The paper studies the identifiability conditions of these models. It also shows that this approach helps disentangle important features, making it possible to improve the use of pre-trained models like CLIP for tasks such as learning with limited examples and adapting to new domains. The method is empirically tested on various real-world datasets.

**Strengths:**

**Originality and significance.** This paper focuses on proposing a more generalized generative framework for studying multimodal settings, which is an interesting and important problem both from a theoretical and empirical perspective. It is clear how this work differs from previous contributions, and the analysis shows new insights.

**Quality.** The paper is well written and the claims are mostly well-substantiated with either proofs or experiments. The authors include an “Insights” section following nearly every theorem, which greatly helps in understanding and building insights.

**Clarity.** The paper is well-organized and clear, with understandable figures and tables.

Table 1 presents interesting results about how the robustness of the results, even in the absence of the assumptions made in the paper

**Weaknesses:**

- Although the authors show three different scenarios of the possible DAGs in Figure 2,  there are fundamental issues in this modeling. There exists a real world that we measure with various sensors, such as the camera (generating images from various views) or textual descriptions of the scene. Hence, for two modalities, each observation requires two types of latent representations: modality-specific representations ($m_x$ and $m_t$) that hold information exclusive to each modality, and a shared representation ($z=z_x=z_t$) that captures common information across both modalities. This eliminates the need to connect $z_x$ and $z_t$ through an undirected edge. Further, this idea represents the DAG shown in Figure 2(a). While the authors justify DAGs shown in Figure 2(b) and Figure 2(c) with examples of text-to-image retrieval and image captioning, these tasks can also be captured as inference on the first DAG i.e. Figure 2(a).

-   Theorem 3.1 is interesting and provides insights into the CLIP objective in an infinite sample setting. However, as also mentioned in the paper, this seems to be a direct extension of Theorem 1 to the multimodal setting in Wang and Isola (2020). I am unsure if it adds a lot of value.

- The claims made regarding the disentanglement capabilities of CLIP seem to be unfounded. Firstly, the authors base their assertions on empirical analysis using only a single dataset, which is insufficient to substantiate claims of disentanglement. Further, the authors state that“the objective of disentangled representations is to learn features that lend themselves to be easily and robustly transferred to downstream tasks”. However, this is more of a consequence of achieving disentangled representations rather than an actual objective. The paper does not provide a clear or rigorous definition of disentanglement (which in itself is an involved question). Moreover, the baselines used for the disentanglement experiments are outdated (from 2018), while there are more recent works in this field.

1.
https://proceedings.neurips.cc/paper_files/paper/2023/hash/6818dcc65fdf3cbd4b05770fb957803e-Abstract-Conference.html

2.
https://proceedings.neurips.cc/paper_files/paper/2023/hash/93e98ddf39a9beb0a97fbbe56a986c80-Abstract-Conference.html

Specifically, the first paper even shows the suboptimality of existing multimodal representation learning approaches such as CLIP. Overall, it appears that the authors use the notion of disentanglement to reinforce their theoretical findings, but their references to it are vague and unsupported. The experimental results presented do not imply “true” disentanglement.

- Minor: Figures 1 and 2 can be placed early on in the text. They are referred to in the introduction. It would be easier to have a main figure that explains the broad concept.

**Questions:**

Please see the weaknesses above.

---

> ### Author Response · Authors · 2024-11-21
> **Response**
>
> We thank Reviewer KBLq for taking the time to review our submission and for recognizing its well-organized structure, clear writing, strong substantiation, and contributions to an interesting and important problem, as well as the new insights it provides. We address the reviewer’s questions and critiques below.
>
> -----
>
> **Q1:** There are fundamental issues for possible DAGs in Figure 2... There exists a real world that we measure with various sensors, such as the camera (generating images from various views) or textual descriptions of the scene... Figure 2(b) and Figure 2(c) can also be captured as inference on the first DAG i.e. Figure 2(a).
>
> **R1:** We agree that the scenario described by the reviewer can be effectively modeled using a shared latent variable across modalities. However, there are real-world examples, such as text-to-image generation (e.g., in datasets like MNIST) and image captioning as seen in datasets like CelebA-Dialog, where the structures depicted in Figures 2(b) and 2(c) are more appropriate. We acknowledge that for many **correlation-based** inference tasks, using slightly mis-specified models can yield results comparable to those obtained with the true model. For instance, employing the model in Figure 2(a) for both learning and inference on data generated by the true data generation processes depicted in Figures 2(b) and 2(c) may produce results similar to those obtained with the corresponding true model. However, the primary advantage of learning a model that aligns with the true process extends beyond **correlation-based** inference tasks. A true causal model enables intervention and counterfactual analysis, which are not possible with a misspecified model.
>
> **Q2:** Theorem 3.1 is interesting and provides insights into the CLIP objective ..... I am unsure if it adds a lot of value.
>
> **R2:** Theorem 3.1 offers an interesting insight into how multimodal contrastive loss is related to addressing inverse problems, as analyzed following the theorem. It is important to note that this insight was not provided in Wang and Isola (2020). Furthermore, Theorem 3.1 serves as a cornerstone for developing Theorems 4.1 and 4.3. In other words, it acts as a bridge connecting the multimodal contrastive loss in Eq. (1) to the results in Theorems 4.1 and 4.3.

---

> > ### Author Response · Authors · 2024-11-21
> >
> > **Q3:** The claims made regarding the disentanglement capabilities of CLIP seem to be unfounded. Firstly,.. using only a single dataset, which is insufficient.... Further, the authors state that“the objective of disentangled representations is..”. However, this is more of a consequence of achieving disentangled representations rather than an actual objective.....
> >
> > **R3:** The claims made regarding the disentanglement capabilities of CLIP are supported by the following points:
> >
> > 1) Theoretical Guarantee: As provided by Theorem 4.1 and discussed in the paragraph that follows it. In summary, the multimodal constrastive loss configuration of CLIP aligns with the assumptions of Theorem 4.1, indicating that CLIP can identify latent coupled variables up to a linear transformation. To extract independent latent variables, a simple application of linear ICA suffices.
> >
> > 2) Simulation Experiments: The results on synthetic data in Table 1(a) support the theoretical findings in Theorem 4.1. Notably, the multimodal constrastive loss configuration of CLIP corresponds to the model scenario where the latent space is a hypersphere and q() follows a vMF (von Mises-Fisher) distribution.
> >
> > 3) Visual Manipulation Results: Based on the identifiability results established for CLIP, we further apply linear ICA to the features learned by CLIP to achieve component-wise identification of the latent variables. The manipulation results on the CelebA dataset validate both our theoretical conclusion and the effectiveness of the proposed method. Demonstrated through experiments on the CelebA dataset.
> >
> > 4) We have provided a clear definition of identifiability/disentanglement in the revision.
> >
> > 5) Few shot Learning and Domain Generalization across various real datasets: We acknowledge that few shot learning and domain generalization may not constitute the most rigorous evaluation criteria for disentanglement. However, establishing a comprehensive and universally accepted evaluation standard is non-trivial, especially given the absence of ground truth in most real-world scenarios [1]. Developing such criteria is beyond the scope of this work. Given this, assessing performance on downstream tasks, such as few shot learning and domain generalization, is a commonly accepted practice within the community [2]. More importantly, this approach is particularly suitable in our context, where we compare the performance of CLIP and CLIP augmented with linear ICA. The only variable in this comparison is the application of linear ICA, ensuring that all other conditions remain consistent and minimizing other influences on the performance in few shot and domain shift tasks. This allows for a meaningful assessment of the impact on transferability and robustness.
> >
> >
> > [1] Locatello, Francesco, et al. "A sober look at the unsupervised learning of disentangled representations and their evaluation." Journal of Machine Learning Research 21.209 (2020): 1-62.
> >
> > [2] Fumero, Marco, et al. "Leveraging sparse and shared feature activations for disentangled representation learning." Advances in Neural Information Processing Systems 36 (2023): 27682-27698.

---

> > > ### Author Response · Authors · 2024-11-21
> > >
> > > **Q4:** it appears that the authors use the notion of disentanglement to reinforce their theoretical findings, but their references to it are vague and unsupported.
> > >
> > > **R4:** We have clarified the notion of disentanglement through differentiability results in the original version. As discussed in the paragraphs following Theorem 4.1, 'leveraging Theorem 4.1, we can show that the minimization of Eq. (5) is capable of identifying the latent variables $\mathbf{z}_x$ $\mathbf{z}_t$ up to a linear transformation, meaning the recovered latent variable $\mathbf{\hat z}_x$ is linearly related to the true $\mathbf{z}_x$ as $\mathbf{\hat z}_x$ = $\mathbf{A}$ $\mathbf{z}_x$ + $\mathbf{c}$, where $\mathbf{A}$ is an orthogonal matrix'. Given this, we can extract $\mathbf{z}_x$ has M−1 independent components using linear ICA. Additionally, for convex bodies, the paragraph following Theorem 4.2 states that Eq. (8) in Theorem 4.2 can identify the latent variables $\mathbf{z}_x$ and $\mathbf{z}_t$ up to a permutation transformation, further supporting the notation used in this work. To clarify these points, we have updated the relevant paragraphs, with the changes highlighted in blue in the new version.
> > >
> > > **Q5:** Specifically, the first paper even shows the suboptimality of existing multimodal representation learning approaches such as CLIP.
> > >
> > > **R5:** When discussing suboptimality, it is crucial to specify the context. The first paper considers a scenario where both shared latent information and task-specific information (distinct from shared latent information) contribute to the task. This is not contradictory to our theoretical results. Our theoretical findings demonstrate that features learned by CLIP identify only the latent coupled variables, i.e., the shared information across modalities. In the context of the first paper, this is insufficient because CLIP does not provide task-specific information. In contrast, the method proposed in the first paper leverages both shared latent information and task-specific information. Thus, it is reasonable to conclude that CLIP is suboptimal in this specific context. More importantly, this observation does not contradict our theoretical results. Instead, it supports them, as our results explicitly show that features learned by CLIP are limited to identifying latent coupled variables.
> > >
> > > **Q6 & R6** Regarding the second paper you mentioned, its primary focus is on proposing a novel contrastive learning framework. In contrast, our work is clearly centered on the theoretical analysis of the traditional multimodal contrastive loss.
> > >
> > > _____
> > >
> > > Dear reviewer KBLq,
> > >
> > > Thank you sincerely for investing your time and consideration into this evaluation process. Our aim in providing this clarification is to furnish valuable insights that address your concerns and contribute to a more nuanced understanding of the contributions and merits of our research. Your thoughtful re-evaluation holds immense significance for us, and we sincerely hope that our clarification proves instrumental in facilitating a comprehensive and constructive reassessment on your part.

---

> > > > ### Comment · Reviewer_KBLq · 2024-11-25
> > > >
> > > > Thank you for your thorough rebuttal. I can see how the causal graph offers valuable contributions to scenarios, such as text-to-image generation and image captioning. Additionally, the rebuttal provides new connections and insights into disentanglement that were previously missing. In light of these changes, I am raising my score to a 6.

---

> > > > > ### Author Response · Authors · 2024-11-25
> > > > >
> > > > > Dear reviewer KBLq,
> > > > >
> > > > > Thank you sincerely for investing your time and, most importantly, for reconsidering our work based on the revised version. We truly appreciate your effort in helping us refine this draft.
> > > > >
> > > > > Best,
> > > > >
> > > > > Authors

---

### Author Response · Authors · 2024-11-21
**General Response**

General Response:

We extend our sincere appreciation to the reviewers for their invaluable feedback. It is gratifying to receive positive remarks across various dimensions:

----------

Theory:

"It is clear how this work differs from previous contributions, and the analysis shows new insights..",
"an interesting and important problem both from a theoretical and empirical perspective" -- Reviewer KBLq


"provides a novel perspective on the latent partial causal model setting.",
"provides identifiability analysis for the provided causal model on both hypersphere and convex body." -- Reviewer B7j4

"The idea of latent partial causal modeling in multi-modal contrastive learning is interesting.",
"the insight that ICA can improve the performance f... is useful and interesting." -- Reviewer 9Eyo


"the paper provides new identifiability guarantees for CRL",
"Some theoretical results on the multi-modal contrastive loss can help one better understand"  -- Reviewer PGzN


----------

Writing:

"the paper is well written and the claims are mostly well-substantiated ." ,
"The paper is well-organized and clear,'" -- Reviewer KBLq

Presentation: 3: good -- Reviewer B7j4

Presentation: 3: good -- Reviewer 9Eyo


----------

Experiments:

" are mostly well-substantiated with either proofs or experiments" ,
"interesting results about how the robustness of the results, even in the absence of the assumptions." -- Reviewer KBLq

"Empirical results on a synthetic setting and CLIP embeddings support the theoretical results." -- Reviewer B7j4

"Empirical results validate the authors' theoretical insights" --Reviewer 9Eyo



----------


In response to specific concerns raised by each reviewer:

In addressing Reviewer KBLq's primary concerns regarding the clarification of disentanglement and the provision of experimental support, we have provided additional elucidation to enhance clarity.

In response to Reviewer B7j4's comments regarding certain technical details, we have provided further clarification.

In response to Reviewer 9Eyo's comments regarding the clarification of non-DAG structures and disentanglement, we have thoroughly addressed these points through expanded explanations.

In response to Reviewer PGzN's comments regarding the presentation, we have thoroughly addressed them by providing clearer motivation and additional details.

We have answered all questions (see more details in the individual responses).

We sincerely thank all reviewers for their thorough evaluations. We are open to further discussion and welcome any additional feedback.

---

### Meta-Review · Area_Chair_ure8 · 2024-12-19

**Metareview:**

This paper was about learning a causal model based off of multimodal data with multimodal contrastive learning. The main result identifies variables in the causal model they pose up to transformations. This analysis surfaces the potential for disentanglement for contrastive learning methods like CLIP. The main concerns among positive reviewers centered around related work, including the relationship to related work that is being explored in self-supervised learning.  These partly clarified in the author results. The negative reviewer would like improvements in the presentation and limitations on theoretical results.

Despite the merits of the general directions, the concerns are enough to make this paper not ready for publication.

**Additional Comments On Reviewer Discussion:**

There was no strong support, though the most negative reviewer noted that there are problems in the presentation and limitations on the theoretical results

---

### Decision · Program_Chairs · 2025-01-22

Reject